# On Using Certified Training towards Empirical Robustness

**Alessandro De Palma**[*]                                                  *alessandro.de-palma@inria.fr*
*Inria, École Normale Supérieure, PSL University, CNRS*

**Serge Durand**[*]                                                            *serge.durand@inria.fr*
*Inria, École Normale Supérieure, PSL University, CNRS*
*Université Paris-Saclay, CEA, List*

**Zakaria Chihani**                                                          *zakaria.chihani@cea.fr*
*Université Paris-Saclay, CEA, List*

**François Terrier**                                                         *francois.terrier@cea.fr*
*Université Paris-Saclay, CEA, List*

**Caterina Urban**                                                           *caterina.urban@inria.fr*
*Inria, École Normale Supérieure, PSL University, CNRS*

**Reviewed on OpenReview:** *https://openreview.net/forum?id=UaaT2fI9DC*

## Abstract

Adversarial training is arguably the most popular way to provide empirical robustness against specific adversarial examples. While variants based on multi-step attacks incur significant computational overhead, single-step variants are vulnerable to a failure mode known as catastrophic overfitting, which hinders their practical utility for large perturbations. A parallel line of work, certified training, has focused on producing networks amenable to formal guarantees of robustness against any possible attack. However, the wide gap between the best-performing empirical and certified defenses has severely limited the applicability of the latter. Inspired by recent developments in certified training, which rely on a combination of adversarial attacks with network over-approximations, and by the connections between local linearity and catastrophic overfitting, we present experimental evidence on the practical utility and limitations of using certified training towards empirical robustness. We show that, when tuned for the purpose, a recent certified training algorithm can prevent catastrophic overfitting on single-step attacks, and that it can bridge the gap to multi-step baselines under appropriate experimental settings. Finally, we present a conceptually simple regularizer for network over-approximations that can achieve similar effects while markedly reducing runtime.

## 1 Introduction

The discovery of adversarial examples (Biggio et al., 2013; Szegedy et al., 2014; Goodfellow et al., 2015), semantic invariant perturbations that induce high-confidence misclassifications in neural networks, has led to the development of a variety of adversarial attacks (Moosavi-Dezfooli et al., 2016; Carlini & Wagner, 2017) and empirical defenses (Papernot et al., 2016; Cisse et al., 2017; Tramèr et al., 2018). Adversarial training (Madry et al., 2018) is undeniably the most successful empirical defense, owing to its efficacy and conceptual simplicity. Employing a robust optimization perspective, Madry et al. (2018) train using the loss incurred at the point returned by a multi-step attack, named PGD. The large cost of PGD-based adversarial training ushered in the development of less expensive techniques (Shafahi et al., 2019). However, single-step attacks (Goodfellow et al., 2015), were shown to suffer from a phenomenon known as Catastrophic

---

[*]Equal contribution.

Overfitting (CO) (Wong et al., 2020), under which they display a vulnerability to multi-step attacks whilst preserving strong robustness to single-step attacks. It was shown that CO can be mitigated at no additional cost through strong noise (de Jorge et al., 2022) for smaller perturbations, yet stronger attack models require the addition of explicit regularizers based on local linearity (Andriushchenko & Flammarion, 2020; Rocamora et al., 2024). Indeed, strong links between lack of local linearity and CO exist (Ortiz-Jimenez et al., 2023).

While empirical robustness to strong attack suites Croce & Hein (2020) is understood to be sufficient for many practical use-cases, a network could still be vulnerable to stronger and unseen attacks, casting doubts over its suitability for safety-critical contexts. In light of this, a variety of techniques to formally verify network robustness have been designed (Katz et al., 2017; Bunel et al., 2018; Ehlers, 2017; Tjeng et al., 2019; Lomuscio & Maganti, 2017). In spite of significant improvements over the last years (Bunel et al., 2020b; Wang et al., 2021; Ferrari et al., 2022), neural network verification techniques still face significant scaling challenges. As a result, a series of works have devised so-called certified training schemes, which produce networks more amenable to formal verification by enforcing small network over-approximations (Gowal et al., 2018; Mirman et al., 2018; Zhang et al., 2020; Wong & Kolter, 2018; Xu et al., 2020). Remarkably, a particularly inexpensive over-approximation based on Interval Bound Propagation (IBP) has emerged as a crucial technique in the area (Jovanović et al., 2022; Mao et al., 2024b). Indeed, the current state-of-the-art in certified training relies on a combination of adversarial attacks and IBP, greatly improving performance over a variety of settings (Müller et al., 2023; Mao et al., 2023). Some of these techniques were formalized into the notion of expressive losses, capable of interpolating between attacks and over-approximations (De Palma et al., 2024b).

On popular vision benchmarks, there is often a wide gap between the best-reported empirical robustness across networks and adversarial training algorithms, and the best-reported certified robustness across certified training schemes (Croce & Hein, 2020; Jovanović et al., 2022; De Palma et al., 2024b). As a result, the latter are typically understood to improve verifiability at the expense of empirical robustness. Intrigued by the versatility of recent certified training algorithms and by the links between local linearity and small network over-approximations, we present a comprehensive empirical study on the applicability of recent certified training techniques towards empirical robustness, leading to the following contributions:

- We systematically show for the first time that, *when explicitly tuned for the purpose*, a multiplicative expressive loss named Exp-IBP (De Palma et al., 2024b) can effectively prevent CO on setups common in the single-step adversarial training literature (§5.1). Under a limited subset of these settings, it can match or overcome the performance of some multi-step adversarial training baselines (PGD-5) while exclusively relying on single-step attacks and enforcing large verifiability.

- We show that the relative empirical robustness of Exp-IBP improves on shallower networks and longer training schedules, demonstrating that pure IBP outperforms a strong multi-step adversarial training baseline (PGD-10) for large perturbation radii on CIFAR-10 (§5.2).

- We present a conceptually simple proxy for the IBP loss, named ForwAbs, which can be computed at the cost of a single network evaluation (§3.3). When used as a regularizer, ForwAbs can prevent CO and improve the robustness of single-step attacks while significantly reducing overhead compared to expressive losses.

We believe our work demonstrates both the potential and the limitations of using current certified training techniques towards empirical robustness, paving the way for further developments in this direction. Code is available at `https://github.com/sergedurand/CertifiedTraining4EmpiricalRobustness`.

## 2 Background

We will use the following notations: $a$ for a scalar, $\mathcal{A}$ for sets, $A$ for matrices, $\boldsymbol{x}$ for a vector and $\boldsymbol{x}_i$ for its $i$-th component. We will furthermore use $[\![a, b]\!]$ for integer ranges, while denoting by $\min_{\boldsymbol{x}} \boldsymbol{f}(\boldsymbol{x})$ the vector given by the entry-wise minimization over the vector function $\boldsymbol{f}$, by $\mathrm{Proj}(\boldsymbol{x}, \mathcal{A})$ the Euclidean projection of $\boldsymbol{x}$ onto $\mathcal{A}$, and by $|A|$ the entry-wise absolute value of $A$. Finally, $\mathbf{1}$ stands for a vector whose entries are all 1.

## 2.1 Adversarial Robustness

Let $\boldsymbol{f_\theta} : \mathbb{R}^d \to \mathbb{R}^k$ be a neural network parameterized by $\boldsymbol{\theta}$. Given an input $\boldsymbol{x} \in \mathbb{R}^d$ and a label $y \in [\![0, k-1]\!]$ from a $k$-way classification dataset $(\boldsymbol{x}, y) \sim \mathcal{D}$, $\boldsymbol{f_\theta}$ is said to be locally robust on $\boldsymbol{x}$ if, given a set of allowed perturbations $\mathcal{C}(\boldsymbol{x}) \subseteq \mathbb{R}^d$:

$$\boldsymbol{x}' \in \mathcal{C}(\boldsymbol{x}) \implies \arg\max_i \boldsymbol{f}_\theta(\boldsymbol{x}')_i = y. \tag{1}$$

In other words, the network is robust if the predicted label for $\boldsymbol{x}$ is invariant to perturbations in $\mathcal{C}(\boldsymbol{x})$. In the following, we will focus on $\ell_\infty$ balls of radius $\epsilon$: $\mathcal{C}(\boldsymbol{x}) = \mathcal{B}_\epsilon(\boldsymbol{x}) := \{\boldsymbol{x}' : ||\boldsymbol{x}' - \boldsymbol{x}||_\infty \leq \epsilon\}$.

Given a surrogate loss $\mathcal{L}$ for classification, typically cross-entropy, the problem of training a network satisfying equation (1) for all $(\boldsymbol{x}, y) \sim \mathcal{D}$, $\boldsymbol{f_\theta}$ can be formulated as a robust optimization problem (Madry et al., 2018):

$$\min_{\boldsymbol{\theta}} \left[ \mathbb{E}_{(\boldsymbol{x},y)\in\mathcal{D}} \mathcal{L}^*(\boldsymbol{f}_\theta, \mathcal{B}_\epsilon(\boldsymbol{x}); y) \right], \quad \text{where} \quad \mathcal{L}^*(\boldsymbol{f}_\theta, \mathcal{B}_\epsilon(\boldsymbol{x}); y) := \max_{\boldsymbol{x}' \in \mathcal{B}_\epsilon(\boldsymbol{x})} \mathcal{L}(\boldsymbol{f}_\theta(\boldsymbol{x}'), y).$$

Computing the worst-case loss $\mathcal{L}^*(\boldsymbol{f}_\theta, \mathcal{B}_\epsilon(\boldsymbol{x}); y)$ exactly is not feasible at training time as, owing to the non-linearity of the network activations, it involves computing the global optimum of a non-concave optimization problem. In the case of piecewise-linear networks, this was shown to be NP-complete (Katz et al., 2017). The worst-case loss is hence commonly replaced by different approximations depending on the training goal.

## 2.2 Training for Empirical Robustness: Adversarial Training

Adversarial attacks are heuristic algorithms, typically based on gradient-based local maximization (Madry et al., 2018) or random search (Andriushchenko et al., 2020), that provide lower bounds to $\mathcal{L}^*(\boldsymbol{f}_\theta, \mathcal{B}_\epsilon(\boldsymbol{x}); y)$. Let us denote by $\boldsymbol{x}_{\mathrm{adv},\mathbb{A}}$ the output of running an attack $\mathbb{A}$ on $\boldsymbol{f}_\theta$ for the perturbation set $\mathcal{B}_\epsilon(\boldsymbol{x})$. A network is deemed to be *empirically robust* to $\mathbb{A}$ for $(\boldsymbol{x}, y)$ if $\arg\max_i \boldsymbol{f}_\theta(\boldsymbol{x}_{\mathrm{adv},\mathbb{A}})_i = y$. When training for empirical robustness, $\mathcal{L}^*(\boldsymbol{f}_\theta, \mathcal{B}_\epsilon(\boldsymbol{x}); y)$ is replaced by the so-called adversarial loss:

$$\mathcal{L}_{\mathrm{adv}}(\boldsymbol{f}_\theta, \mathcal{B}_\epsilon(\boldsymbol{x}); y) := \mathcal{L}(\boldsymbol{f}_\theta(\boldsymbol{x}_{\mathrm{adv},\mathbb{A}}), y),$$

which is a lower bound to $\mathcal{L}^*(\boldsymbol{f}_\theta, \mathcal{B}_\epsilon(\boldsymbol{x}); y)$. One of the most popular and effective adversarial training algorithms employs the PGD attack (Madry et al., 2018) at training time. Starting from a random uniform input in $\mathcal{B}_\epsilon(\boldsymbol{x})$, PGD iteratively steps using the sign of the network input gradient $\mathrm{sign}\,(\nabla_{\boldsymbol{x}}\mathcal{L}(\boldsymbol{f}_\theta(\boldsymbol{x}), \boldsymbol{y}))$, projecting back to $\mathcal{B}_\epsilon(\boldsymbol{x})$ after every iteration. Owing to the large cost of training with multi-step PGD, a series of works have focused on designing effective single-step alternatives (Shafahi et al., 2019; Wong et al., 2020; de Jorge et al., 2022). Among them is FGSM (Goodfellow et al., 2015), which systematically lands on a corner of the perturbation space: $\boldsymbol{x}_{\mathrm{adv},\mathrm{FGSM}} = \boldsymbol{x} + \epsilon\, \mathrm{sign}\,(\nabla_{\boldsymbol{x}}\mathcal{L}(\boldsymbol{f}_\theta(\boldsymbol{x}), \boldsymbol{y}))$. FGSM was shown to suffer from a failure mode known as *catastrophic overfitting* (Wong et al., 2020), under which the network's robustness to PGD attacks rapidly drops to $\approx 0\%$. Wong et al. (2020) empirically demonstrate that catastrophic overfitting can be prevented at moderate $\epsilon$ values if using a random uniform input in $\mathcal{B}_\epsilon(\boldsymbol{x})$ as a FGSM starting point and increasing the step size from $\eta = \epsilon$ to $\eta = 1.25\epsilon$ (RS-FGSM). de Jorge et al. (2022) propose to sample uniformly from a larger set than the target model (typically, $\mathcal{B}_{2\epsilon}(\boldsymbol{x})$), using a step size of $\eta = \epsilon$, and by removing RS-FGSM's projection onto $\mathcal{B}_\epsilon(\boldsymbol{x})$ after the step. The resulting attack, named N-FGSM, attains non-negligible PGD robustness at larger $\epsilon$ without incurring any additional cost with respect to RS-FGSM, and acts as an implicit loss regularization (de Jorge et al., 2022). Nevertheless, explicit regularizers, often enforcing local linearity (Andriushchenko & Flammarion, 2020; Rocamora et al., 2024) are needed to further mitigate CO against stronger attacks.

## 2.3 Training for Verified Robustness: Certified Training

As described in §2.2, the notion of empirical robustness is specific to a given attack $\mathbb{A}$. However, $\boldsymbol{f}_\theta(\boldsymbol{x}_{\mathrm{adv},\mathbb{A}})_i = y$ does not necessarily imply that equation (1) holds, as the network may still be vulnerable to a more effective or unseen attack $\mathbb{A}'$. Formal verification provides a stronger, attack-independent, notion of robustness. A network $\boldsymbol{f}_\theta$ is said to be *certifiably robust* for $(\boldsymbol{x}, y)$ if and only if the difference

between the ground truth logit $\boldsymbol{f_\theta}(\boldsymbol{x'})_y$ and the other logits is positive for all $\boldsymbol{x'} \in \mathcal{B}_\epsilon(\boldsymbol{x})$:

$$\min_i \left\{ \boldsymbol{z}_{\boldsymbol{f_\theta}}^{\mathcal{B}_\epsilon(\boldsymbol{x}),y} := \min_{\boldsymbol{x'} \in \mathcal{B}_\epsilon(\boldsymbol{x})} \left[ \boldsymbol{z}_{\boldsymbol{f_\theta}}(\boldsymbol{x},y) := (\boldsymbol{f_\theta}(\boldsymbol{x'})_y - \boldsymbol{f_\theta}(\boldsymbol{x'})) \right] \right\}_i \geq 0. \tag{2}$$

Assuming the use of a loss function $\mathcal{L}$ that is translation invariant with respect to the network output (Wong et al., 2020), such as cross-entropy, then $\mathcal{L}^*(\boldsymbol{f_\theta}, \mathcal{B}_\epsilon(\boldsymbol{x}); y) = \mathcal{L}\left(-\boldsymbol{z}_{\boldsymbol{f_\theta}}^{\mathcal{B}_\epsilon(\boldsymbol{x}),y}, y\right)$. As for $\mathcal{L}^*(\boldsymbol{f_\theta}, \mathcal{B}_\epsilon(\boldsymbol{x}); y)$ (see §2.1), computing the exact optimal logit differences $\boldsymbol{z}_{\boldsymbol{f_\theta}}^{\mathcal{B}_\epsilon(\boldsymbol{x}),y}$ is computationally infeasible. However, given lower bounds $\underline{\boldsymbol{z}}_{\boldsymbol{f_\theta}}^{\mathcal{B}_\epsilon(\boldsymbol{x}),y} \leq \boldsymbol{z}_{\boldsymbol{f_\theta}}^{\mathcal{B}_\epsilon(\boldsymbol{x}),y}$ to the optimal logit differences, we can conclude that $\boldsymbol{f_\theta}$ is certifiably robust if $\min_i \left(\underline{\boldsymbol{z}}_{\boldsymbol{f_\theta}}^{\mathcal{B}_\epsilon(\boldsymbol{x}),y}\right)_i \geq 0$. We will henceforth call a NN verifier any method to compute the above lower bounds. Given an efficient verifier, and assuming that the loss is monotonically increasing for all the network outputs except the one associated with the ground truth, networks can be trained for certified robustness by using the loss induced by the lower bounds to the logit differences (Wong & Kolter, 2018; Zhang et al., 2020), which upper bounds the worst-case loss:

$$\mathcal{L}_{\text{ver}}(\boldsymbol{f_\theta}, \mathcal{B}_\epsilon(\boldsymbol{x}); y) := \mathcal{L}\left(-\underline{\boldsymbol{z}}_{\boldsymbol{f_\theta}}^{\mathcal{B}_\epsilon(\boldsymbol{x}),y}, y\right) \geq \mathcal{L}^*(\boldsymbol{f_\theta}, \mathcal{B}_\epsilon(\boldsymbol{x}); y). \tag{3}$$

Networks trained via $\mathcal{L}_{\text{ver}}(\boldsymbol{f_\theta}, \mathcal{B}_\epsilon(\boldsymbol{x}); y)$ feature tight $\underline{\boldsymbol{z}}_{\boldsymbol{f_\theta}}^{\mathcal{B}_\epsilon(\boldsymbol{x}),y}$ lower bounds by design. We will focus on the least expensive bounding algorithm: IBP (Mirman et al., 2018; Gowal et al., 2018). When used for training, IBP outperforms tighter relaxations as it results in better-behaved loss functions (Jovanović et al., 2022).

### 2.3.1 IBP

IBP is a method to over-approximate neural network outputs based on the application of interval arithmetics (Sunaga, 1958; Moore, 1966) onto the functions composing the neural architecture. As such, it can be easily applied onto general computational graphs (Xu et al., 2020). For ease of presentation, let us assume that $\boldsymbol{f_\theta}$ is a $n$-layer feed-forward neural network, with layer $j \in [\![1, n-1]\!]$ composed of an affine operation followed by a monotonically increasing element-wise activation function $\boldsymbol{\sigma}$ such as ReLU, $\boldsymbol{x}^j = \boldsymbol{\sigma}\left(W^i \boldsymbol{x}^{j-1} + \boldsymbol{b}^i\right)$, and a final affine layer: $\boldsymbol{f_\theta} = \left(W^n \boldsymbol{x}^{n-1} + \boldsymbol{b}^n\right)$. In this case, we can compose the logit differences and the last layer into a single affine layer: $\boldsymbol{z}_{\boldsymbol{f_\theta}}(\boldsymbol{x}, y) = \left(\tilde{W}^n \boldsymbol{x}^{n-1} + \tilde{\boldsymbol{b}}^n\right)$. Then, for perturbation $\mathcal{B}_\epsilon(\boldsymbol{x}^0)$, IBP computes $\underline{\boldsymbol{z}}_{\boldsymbol{f_\theta}}^{\mathcal{B}_\epsilon(\boldsymbol{x}),y}$ through the following procedure, which iteratively derives lower and upper bounds $\hat{\boldsymbol{l}}^k$ and $\hat{\boldsymbol{u}}^k$ to the outputs of the $k$-th affine layer:

$$\underline{\boldsymbol{z}}_{\boldsymbol{f_\theta}}^{\mathcal{B}_\epsilon(\boldsymbol{x}),y} = \tfrac{1}{2}\tilde{W}^n\left(\hat{\boldsymbol{l}}^{n-1} + \hat{\boldsymbol{u}}^{n-1}\right) - \tfrac{1}{2}|\tilde{W}^n|\left(\hat{\boldsymbol{u}}^{n-1} - \hat{\boldsymbol{l}}^{n-1}\right) + \tilde{\boldsymbol{b}}^n, \quad \text{where:} \begin{bmatrix} \hat{\boldsymbol{l}}^1 = W^1 \boldsymbol{x}^0 - \epsilon \left|W^1\right| \mathbf{1} + \boldsymbol{b}^1 \\ \hat{\boldsymbol{u}}^1 = W^1 \boldsymbol{x}^0 + \epsilon \left|W^1\right| \mathbf{1} + \boldsymbol{b}^1 \end{bmatrix},$$

$$\begin{bmatrix} \hat{\boldsymbol{l}}^j = \tfrac{1}{2}W^j \left(\boldsymbol{\sigma}\left(\hat{\boldsymbol{l}}^{j-1}\right) + \boldsymbol{\sigma}\left(\hat{\boldsymbol{u}}^{j-1}\right)\right) - \tfrac{1}{2}|W^j|\left(\boldsymbol{\sigma}\left(\hat{\boldsymbol{u}}^{j-1}\right) - \boldsymbol{\sigma}\left(\hat{\boldsymbol{l}}^{j-1}\right)\right) + \boldsymbol{b}_j \\ \hat{\boldsymbol{u}}^j = \tfrac{1}{2}W^j \left(\boldsymbol{\sigma}\left(\hat{\boldsymbol{l}}^{j-1}\right) + \boldsymbol{\sigma}\left(\hat{\boldsymbol{u}}^{j-1}\right)\right) + \tfrac{1}{2}|W^j|\left(\boldsymbol{\sigma}\left(\hat{\boldsymbol{u}}^{j-1}\right) - \boldsymbol{\sigma}\left(\hat{\boldsymbol{l}}^{j-1}\right)\right) + \boldsymbol{b}_j \end{bmatrix} \forall\, j \in [\![2, n-1]\!]. \tag{4}$$

We will henceforth write $\boldsymbol{l}_{\boldsymbol{f_\theta}}^{\mathcal{B}_\epsilon(\boldsymbol{x}),y}$ for the lower bounds to the logit differences obtained through equation (4), and $\mathcal{L}_{\text{IBP}}(\boldsymbol{f_\theta}, \mathcal{B}_\epsilon(\boldsymbol{x}); y) := \mathcal{L}\left(-\boldsymbol{l}_{\boldsymbol{f_\theta}}^{\mathcal{B}_\epsilon(\boldsymbol{x}),y}, y\right)$ for the associated loss.

### 2.3.2 Hybrid Methods

A network trained via the IBP loss $\mathcal{L}_{\text{IBP}}(\boldsymbol{f_\theta}, \mathcal{B}_\epsilon(\boldsymbol{x}); y)$ can be easily verified to be robust using the same bounds employed for training. However, the relative looseness of the over-approximation from equation (4) results in a significant decrease in standard performance. For ReLU networks, which are the main focus of this work, the availability of more effective verifiers based on branch-and-bound (Henriksen & Lomuscio, 2021; Wang et al., 2021; Ferrari et al., 2022) post-training has motivated the development of alternative techniques. A series of methods have shown that combination of adversarial training with network over-approximations (Balunovic & Vechev, 2020; Fan & Li, 2021; De Palma et al., 2022; Mao et al., 2023) can yield strong verifiability via branch-and-bound while reducing the impact on standard performance. In

particular, Müller et al. (2023) proposed to compute the IBP loss over a tunable subset of $\mathcal{B}_\epsilon(\boldsymbol{x})$ containing adversarial examples, yielding favorable trade-offs between standard performance and certified robustness via branch-and-bound-based methods. De Palma et al. (2024b) introduced the notion of *loss expressivity*, defined as the ability of a loss function to spawn a continuous range of trade-offs between the adversarial and the IBP losses, showing that expressive losses obtained through convex combinations between an adversarial and an over-approximation component lead to state-of-the-art certified robustness.

While past work in certified training has designed and deployed these algorithms to maximize certified robustness, empirical robustness through strong attacks (Croce & Hein, 2020) remains the most adopted metric owing to the lack of scalability and to the looseness of network verifiers. Noting the potential versatility of recent hybrid certified training schemes, this work systematically investigates whether they can be employed towards improving empirical robustness.

## 3 Certified Training for Empirical Robustness

Recent certified training techniques (§2.3.2) have demonstrated that the ability to precisely control the tightness of network over-approximations while preserving an adversarial training component is crucial to maximize certified robustness across experimental setups. We hypothesize that this versatility can be alternatively leveraged towards improving the empirical robustness of the adversarial training schemes on top of which they are applied, providing experimental evidence in §5. This section details the certified training algorithms employed in the experimental study. We begin with a motivating example (§3.1), then study the qualitative behavior of existing methods in settings of interest (§3.2), and conclude by presenting a conceptually simple regularizer designed to mimic the effect of IBP-based training while reducing the associated overhead (§3.3).

### 3.1 Motivating Example

In order to motivate our empirical study, we show that multi-step attacks, which typically exhibit superior robustness at the expense of training time, are associated to smaller network over-approximations, as measured by the IBP loss. Figure 1, which is associated to the experimental setup from table 1, shows that the IBP loss indeed decreases with the number of attack steps. Furthermore, it shows two distinct qualitative trends: for N-FGSM, which is prone to CO (causing the large experimental variability), the IBP loss increases with the perturbation radius. On the other hand, $\mathcal{L}_{\mathrm{IBP}}(\boldsymbol{f}_\theta, \mathcal{B}_\epsilon(\boldsymbol{x}); y)$ decreases with $\epsilon$ for PGD-5 and PGD-10. Indeed, multi-step attacks yield networks that are more locally linear compared to single-

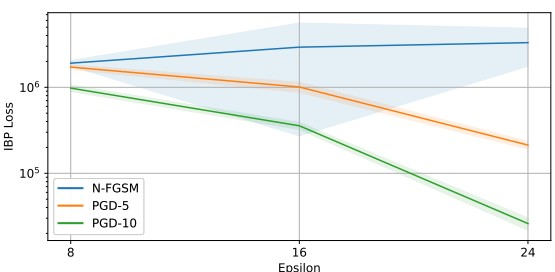

Figure 1: IBP loss of adversarial training schemes on CIFAR-10, setup from table 1.

step attacks (Andriushchenko & Flammarion, 2020; Rocamora et al., 2024; Ortiz-Jimenez et al., 2023), which in turn results in tighter $\underline{\mathbf{z}}_{\boldsymbol{f_\theta}}^{\mathcal{B}_\epsilon(\boldsymbol{x}),y}$ bounds and hence smaller network over-approximations. In this work, we investigate whether directly controlling $\underline{\mathbf{z}}_{\boldsymbol{f_\theta}}^{\mathcal{B}_\epsilon(\boldsymbol{x}),y}$ via certified training can benefit empirical robustness.

### 3.2 Expressive Losses

For a fixed verifier and attack algorithm, an expressive loss $\mathcal{L}_\alpha(\boldsymbol{f}_\theta, \mathcal{B}_\epsilon(\boldsymbol{x}); y)$ can interpolate from the adversarial loss $\mathcal{L}_{\mathrm{adv}}(\boldsymbol{f}_\theta, \mathcal{B}_\epsilon(\boldsymbol{x}); y)$ to the worst-case loss over-approximation $\mathcal{L}_{\mathrm{ver}}(\boldsymbol{f}_\theta, \mathcal{B}_\epsilon(\boldsymbol{x}); y)$ by continuously increasing a parameter $\alpha$, the over-approximation coefficient, from 0 to 1. Formally (De Palma et al., 2024b, Definition 3.1), assuming that IBP is the verifier of choice: $\mathcal{L}_\alpha(\boldsymbol{f}_\theta, \mathcal{B}_\epsilon(\boldsymbol{x}); y)$ is monotonically increasing and continuous with respect to $\alpha$, $\mathcal{L}_0(\boldsymbol{f}_\theta, \mathcal{B}_\epsilon(\boldsymbol{x}); y) = \mathcal{L}_{\mathrm{adv}}(\boldsymbol{f}_\theta, \mathcal{B}_\epsilon(\boldsymbol{x}); y)$, $\mathcal{L}_1(\boldsymbol{f}_\theta, \mathcal{B}_\epsilon(\boldsymbol{x}); y) = \mathcal{L}_{\mathrm{IBP}}(\boldsymbol{f}_\theta, \mathcal{B}_\epsilon(\boldsymbol{x}); y)$, and:

$$\mathcal{L}_{\mathrm{adv}}(\boldsymbol{f}_\theta, \mathcal{B}_\epsilon(\boldsymbol{x}); y) \leq \mathcal{L}_\alpha(\boldsymbol{f}_\theta, \mathcal{B}_\epsilon(\boldsymbol{x}); y) \leq \mathcal{L}_{\mathrm{IBP}}(\boldsymbol{f}_\theta, \mathcal{B}_\epsilon(\boldsymbol{x}); y) \ \forall \ \alpha \in [0, 1].$$

As shown by De Palma et al. (2024b) expressive losses include SABR (Müller et al., 2023) and three loss functions defined through convex combinations: CC-IBP, MTL-IBP and Exp-IBP. We found MTL-IBP,

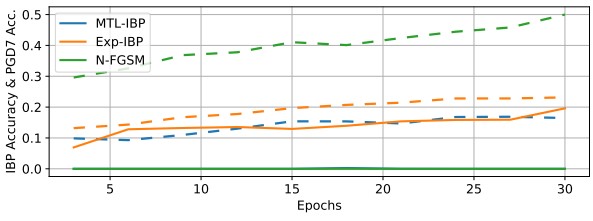
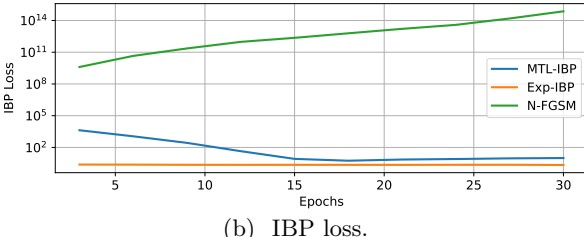

(a) Certified accuracy via IBP bounds (solid lines), empirical accuracy under PGD-7 attacks (dashed).

(b) IBP loss.

Figure 2: IBP certified robustness attained by MTL-IBP and Exp-IBP on the `PreActResNet18` training setup from de Jorge et al. (2022). Validation results on CIFAR-10 under perturbations of $\epsilon = {}^8/_{255}$.

CC-IBP and SABR to display similar behaviors in the settings of interest: we refer the interested reader to appendix A for additional details. For simplicity and brevity, we will hence focus on MTL-IBP and Exp-IBP in the remainder of this work. The former consists of a convex combination of the adversarial and IBP losses. Exp-IBP, instead, performs the loss convex combination in a logarithmic space.

**Definition 3.1** *The MTL-IBP and Exp-IBP losses are defined as follows:*

$$
\begin{aligned}
\mathcal{L}_\alpha^{MTL\text{-}IBP}(\boldsymbol{f_\theta}, \mathcal{B}_\epsilon(\boldsymbol{x}); y) &:= (1-\alpha)\mathcal{L}_{adv}(\boldsymbol{f_\theta}, \mathcal{B}_\epsilon(\boldsymbol{x}); y) + \alpha\mathcal{L}_{IBP}(\boldsymbol{f_\theta}, \mathcal{B}_\epsilon(\boldsymbol{x}); y), \\
\mathcal{L}_\alpha^{Exp\text{-}IBP}(\boldsymbol{f_\theta}, \mathcal{B}_\epsilon(\boldsymbol{x}); y) &:= \mathcal{L}_{adv}(\boldsymbol{f_\theta}, \mathcal{B}_\epsilon(\boldsymbol{x}); y)^{(1-\alpha)}\mathcal{L}_{IBP}(\boldsymbol{f_\theta}, \mathcal{B}_\epsilon(\boldsymbol{x}); y)^\alpha.
\end{aligned}
\tag{5}
$$

**Expressive losses for certified robustness** In order to maximize certified accuracy (defined as the share of inputs verified to be robust) and stabilize training, previous work (Müller et al., 2023; De Palma et al., 2024b) deployed the losses in equation (5) under a specific set of conditions. First, shallow architectures, such as the popular 7-layer `CNN-7` (Shi et al., 2021; Müller et al., 2023; Mao et al., 2023; De Palma et al., 2024b) (see appendix), which, aided by specialized initialization and regularization (Shi et al., 2021), contain the growth of the IBP bounds over consecutive layers. Second, relatively long training schedules featuring gradient clipping, a warm-up phase of standard training, and a ramp-up phase during which the perturbation radius used to compute both the attack and the IBP bounds is gradually increased from 0 to the target value $\epsilon$. Finally, relatively large values of $\alpha$ to reduce the magnitude of the bounds $\underline{\mathbf{z}}_{\boldsymbol{f_\theta}}^{\mathcal{B}_\epsilon(\boldsymbol{x}),y}$ on the logit differences. In this context, all the expressive losses from equation (5) display a similar qualitative behavior, attaining roughly the same certified robustness (De Palma et al., 2024b).

**Qualitative differences in adversarial training setups** In adversarial training, overhead is often a major concern (Shafahi et al., 2019). Indeed, works on single-step adversarial training typically rely on deeper networks and shorter training schedules to maximize empirical robustness while minimizing runtimes. A common setting, which we reproduce in §5.1 because of its prevalence in the catastrophic overfitting literature, involves a `PreactResNet18` (He et al., 2016) trained with a cyclic learning rate and without any gradient clipping, ramp-up (except on SVHN) nor warm-up (Andriushchenko & Flammarion, 2020; Wong et al., 2020; de Jorge et al., 2022). Owing to the exploding IBP loss at initialization (see appendix D.6) and

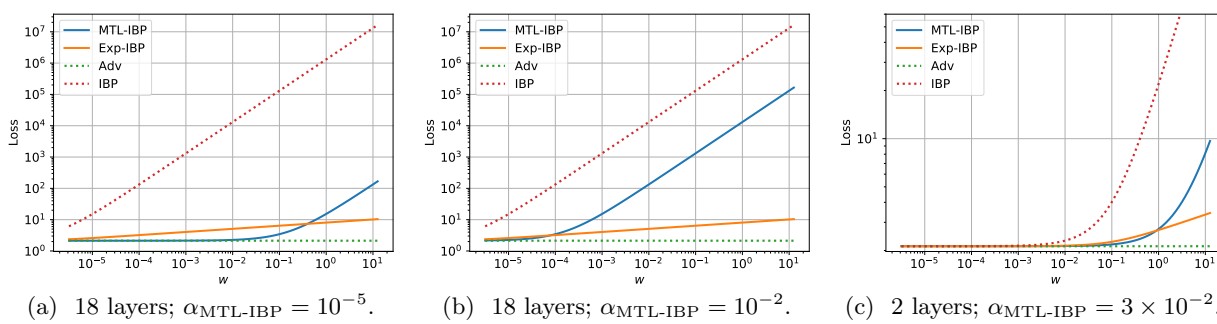

(a) 18 layers; $\alpha_{\text{MTL-IBP}} = 10^{-5}$.

(b) 18 layers; $\alpha_{\text{MTL-IBP}} = 10^{-2}$.

(c) 2 layers; $\alpha_{\text{MTL-IBP}} = 3 \times 10^{-2}$.

Figure 3: Sensitivity of the expressive losses on a toy network of varying depth, with $\alpha_{\text{Exp-IBP}} = 10^{-1}$.

to the specific training hyper-parameters, pure IBP training is not directly applicable to this setup. In order to study the behavior of expressive losses in this context, we tuned MTL-IBP and Exp-IBP to maximize IBP-based verifiability on CIFAR-10 under perturbations of radius $\epsilon = {}^8/_{255}$, using N-FGSM to generate $\boldsymbol{x}_{\mathrm{adv},\mathbb{A}}$. Figure 2 shows that MTL-IBP is unable to attain non-negligible certified accuracy via IBP in this context. On the other hand, Exp-IBP reaches almost 20% IBP certified accuracy, highlighting a significant qualitative difference in behavior. In order to gain further insight on this, we plot the sensitivity of the two expressive losses on two toy networks where the magnitude of the IBP bounds is controlled through a scalar parameter $w$ (see appendix B.1). Figure 3 shows that on the deeper network the MTL-IBP loss either flattens onto $\mathcal{L}_{\mathrm{adv}}(\boldsymbol{f}_\theta, \mathcal{B}_\epsilon(\boldsymbol{x}); y)$ at large IBP loss values (small $\alpha$), or explodes with the IBP bounds, resulting in unstable training behavior (larger $\alpha$). On the other hand, Exp-IBP can tuned so as to be able to drive $\mathcal{L}_{\mathrm{IBP}}(\boldsymbol{f}_\theta, \mathcal{B}_\epsilon(\boldsymbol{x}); y)$ to very small values without exploding at larger $w$ values. On the shallow network, instead, for which the IBP bounds are significantly smaller, MTL-IBP can be tuned to display roughly the same behavior as Exp-IBP for smaller $\mathcal{L}_{\mathrm{IBP}}(\boldsymbol{f}_\theta, \mathcal{B}_\epsilon(\boldsymbol{x}); y)$ without taking overly large loss values within the considered parameter range. We believe this explains their homogeneous behavior in training setups designed to prevent large IBP bounds, such as those typically employed in the certified training literature.

**Maximizing empirical accuracy**  As seen in figure 2, expressive losses can pay a large price in empirical accuracy compared to $\alpha = 0$ (pure N-FGSM, in this case) when $\alpha$ is tuned to tighten the IBP bounds. One may hence conclude that certified robustness is at odds with empirical accuracy. In section §5 we instead show that, when $\alpha$ is tuned for the purpose, expressive losses can result in increased empirical robustness.

### 3.3  ForwAbs

Expressive losses require the computation of IBP bounds $\boldsymbol{l}_{\boldsymbol{f}_\theta}^{\mathcal{B}_\epsilon(\boldsymbol{x}),y}$ using the procedure in equation (4), whose cost roughly corresponds to two network evaluations (also called forward passes): one using the original network, the other employing the absolute value of the network weights. This overhead on top of adversarial training is negligible only when $\boldsymbol{x}_{\mathrm{adv},\mathbb{A}}$ are computed via multi-step attacks. We here study less expensive yet conceptually simple ways to control the IBP bounds when single-step attacks are instead employed.

The gap between the lower and upper bounds after the $k$-th affine layer, which we denote by $\boldsymbol{\delta}^k := (\hat{\boldsymbol{u}}^k - \hat{\boldsymbol{l}}^k)$, can be employed as a measure of the tightness of the IBP bounds (Shi et al., 2021).

For ReLU networks, $\boldsymbol{\delta}^k$ can be upper bound by $|W^k|\boldsymbol{\delta}^{k-1} \geq |W^k|(\sigma(\hat{\boldsymbol{u}}^{k-1}) - \sigma(\hat{\boldsymbol{l}}^{k-1}))$. This upper bound coincides with $\boldsymbol{\delta}^k$ in the case of Deep Linear Networks (DLNs), for which $\sigma(\boldsymbol{a}) = \boldsymbol{a}$, or if the ReLUs are passing ($\hat{\boldsymbol{l}}^k \geq 0$). We propose to employ $\bar{\boldsymbol{\delta}}^n$, which we define as the (looser) upper bound to $\boldsymbol{\delta}^n$ obtained through repeated use of the $|W^k|\boldsymbol{\delta}^{k-1}$ bound, as a regularizer for the IBP loss. The $\bar{\boldsymbol{\delta}}^n$ term can be computed at the cost of a single forward pass through an auxiliary

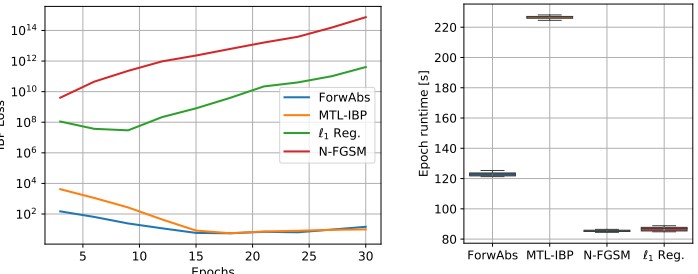

Figure 4:  IBP loss over epochs (*left*), box plots (10 runs) for the training time of an epoch (*right*), setup as figure 2.

network with $\sigma(\boldsymbol{a}) = \boldsymbol{a}$ where each affine layer $\mathbf{g}^k(\boldsymbol{a}) = W^k\boldsymbol{a} + \boldsymbol{b}$ is replaced by $\mathbf{h}^k(\boldsymbol{a}) = |W|^k\boldsymbol{a}$. We call the resulting method ForwAbs (*Forw*ard pass in *Abs*olute value).

**Definition 3.2** *The ForwAbs loss takes the following form:*

$$\mathcal{L}_\lambda^{ForwAbs}(\boldsymbol{f}_\theta, \mathcal{B}_\epsilon(\boldsymbol{x}); y) := \mathcal{L}_{adv}(\boldsymbol{f}_\theta, \mathcal{B}_\epsilon(\boldsymbol{x}); y) + \lambda\left(\mathbf{1}^T\bar{\boldsymbol{\delta}}^n\right),$$
$$where:\quad \bar{\boldsymbol{\delta}}^1 = \boldsymbol{\delta}^1 = 2\epsilon|W^1|\mathbf{1}, \quad \bar{\boldsymbol{\delta}}^k = |W^k|\bar{\boldsymbol{\delta}}^{k-1} \ \forall \ k \in [\![2, n]\!].$$
(6)

In order to demonstrate the empirical correlation between the ForwAbs term $\bar{\boldsymbol{\delta}}^n$ and the IBP bounds $\boldsymbol{l}_{\boldsymbol{f}_\theta}^{\mathcal{B}_\epsilon(\boldsymbol{x}),y}$, figure 4 presents the results of a ForwAbs-trained network, where the $\lambda$ coefficient is tuned to minimize the IBP bounds in the same setup as figure 2. In spite of its simplicity and of its significantly

smaller runtime overhead, ForwAbs can drive the IBP loss to roughly the same values attained by MTL-IBP. Furthermore, the final $\mathcal{L}_{\text{IBP}}(\boldsymbol{f}_\theta, \mathcal{B}_\epsilon(\boldsymbol{x}); y)$ attained by ForwAbs is significantly lower than those associated to N-FGSM. Strong $\ell_1$ regularization over the weights (with regularization coefficient $\lambda_{\ell_1} = 0.04$) fails to achieve a comparable effect, demonstrating that the behavior of ForwAbs is not merely a consequence of small $\ell_1$ norms of the network weights.

## 4 Related Work

We here provide an overview of closely-related work across certified and adversarial training.

### 4.1 Neural Network Verification and Certified Training

As outlined in §2.3, a neural network verifier computes lower bounds to the worst-case logit differences $\boldsymbol{z}_{\boldsymbol{f}_\theta}^{\mathcal{B}_\epsilon(\boldsymbol{x}), y}$. Given its centrality to state-of-the-art certified training schemes (§2.3.2), we have focused on the inexpensive IBP algorithm (§2.3.1). However, a significant number of verifiers exist, for which tighter lower bounds are typically associated with larger overhead. One line of work, inspired by abstract interpretation (Cousot & Cousot, 1977) builds abstract domains over-approximating the neural network (Singh et al., 2018; 2019b). Lower bounds can be also obtained by solving linear programs replacing network non-linearities with convex relaxations (Ehlers, 2017; Anderson et al., 2020; Tjandraatmadja et al., 2020; Singh et al., 2019a), potentially through dual formulations (Wong & Kolter, 2018; Dvijotham et al., 2018; Bunel et al., 2020a; De Palma et al., 2021; 2024a). Alternatively, a series of fast verifiers relies on back-propagating linear bounds through the network (Zhang et al., 2018; Xu et al., 2021; Wang et al., 2021). Tight bounds can be plugged in within a branch-and-bound framework (Bunel et al., 2018; 2020b) to yield a verifier that solve the verification problem in equation (2) exactly for piece-wise linear networks, with state-of-the-art complete verifiers often relying on a combination of linear bound propagation with dual bounds (Ferrari et al., 2022; Zhang et al., 2022b). Earlier certified training schemes (Gowal et al., 2018; Wong & Kolter, 2018; Zhang et al., 2020; Mirman et al., 2018; Shi et al., 2021) typically rely on linear bound propagation or IBP to create a loss function for certified training using equation (3). (Boopathy et al., 2021) present SingleProp: a regularizer which, computed at the cost of a single network evaluation and applied on top of the clean network loss, trades certified robustness for computational efficiency. As described in §2.3.2, many later schemes combine IBP bounds from equation (4) with adversarial training, attaining better certified robustness when using branch-and-bound post-training. We here rely on expressive losses (§3.2) owing to their versatility and state-of-the-art performance, and on ForwAbs (§3.3), a conceptually-simpler alternative to SingleProp which we add on adversarial losses to enhance empirical robustness. Finally, while we focus on $\ell_\infty$ certified training schemes applicable to ReLU networks and based on deterministic lower bounds to $\boldsymbol{z}_{\boldsymbol{f}_\theta}^{\mathcal{B}_\epsilon(\boldsymbol{x}), y}$, alternative techniques include randomized methods (Cohen et al., 2019), Lipschitz regularization (Huang et al., 2021), and ad-hoc architectures (Xu et al., 2022; Zhang et al., 2022a).

### 4.2 Adversarial Training and Catastrophic Overfitting

For simplicity, this work focuses on the widely popular multi-step PGD training (Madry et al., 2018). Nevertheless, multiple variants have been designed to improve its performance (Zhang et al., 2019; Wang et al., 2019; Liu et al., 2021; Xu et al., 2023). Multi-step PGD is coupled with random search within AutoAttack (Croce & Hein, 2020), a strong attack suite that has become the standard way to reliably evaluate empirical robustness. It was noted that adversarial training is prone to overfitting with respect to empirical robustness, a phenomenon known as robust overfitting (Rice et al., 2020), with specialized remedies including smoothing methods (Chen et al., 2020) and techniques linked to a game-theoretical interpretation of the phenomenon (Wang et al., 2024). In this work, we are rather concerned with catastrophic overfitting, a thoroughly-studied (Vivek & Babu, 2020; Park & Lee, 2021; Li et al., 2022; Tsiligkaridis & Roberts, 2022) failure mode associated with one-step adversarial training, and which can be reduced without any additional cost by the appropriate addition of noise to the attacks (Wong et al., 2020; de Jorge et al., 2022): see §2.1. Other techniques include explicit regularizers on the loss smoothness (Sriramanan et al., 2020; 2021), dynamically varying the attack step size (Kim et al., 2021), or selectively zeroing some coordinates of the attack step (Golgooni et al., 2023): these are all outperformed by the noise-based N-FGSM (de Jorge et al.,

2022). Regrettably, even N-FGSM suffer from catastrophic overfitting at larger perturbation radii. Better robustness at large $\epsilon$, albeit at additional cost, can be obtained by promoting the local linearity of the network, which is closely-linked with catastrophic overfitting (Ortiz-Jimenez et al., 2023): GradAlign (Andriushchenko & Flammarion, 2020) aligns the input gradients of clean and randomly perturbed samples, ELLE (Rocamora et al., 2024) by enforcing a simple condition on two random samples from the perturbation set and their convex combination. Recent alternatives to attain large-$\epsilon$ robustness include hindering the generation of adversarial examples with a larger loss than the clean input (Lin et al., 2023), or the use of adaptive weight perturbations (Lin et al., 2024). In section 5.1 we systematically investigate, to the best of our knowledge for the first time, whether catastrophic overfitting on settings from the above literature can be instead mitigated through certified training techniques.

### 4.3 Empirical Robustness of Certified Training

Previous works have reported, explicitly or implicitly, on the empirical robustness of certified training schemes deployed to maximize certified robustness. Except for De Bartolomeis et al. (2023), whose main conclusions focus on the inferior empirical robustness of the certified training schemes they considered compared to a multi-step baseline, complying with the folklore in the area, empirical robustness was not the main focus of these works.

Gowal et al. (2018) showed that IBP can attain larger empirical robustness than adversarial training baselines on relatively shallow networks for MNIST with $\epsilon \in \{0.3, 0.4\}$ and for CIFAR-10 with $\epsilon = 8/255$ when using long training schedules. In contrast with their main claims, similar results, albeit using inconsistent training schedules and optimizers across methods, are presented in De Bartolomeis et al. (2023, appendix B). A very recent work (Mao et al., 2024a) reports further analogous results, showing that a series of certified training schemes tuned to maximize certified accuracy outperform multi-step PGD on MNIST with $\epsilon = 0.3$ and CIFAR-10 with $\epsilon = 8/255$. Nevertheless, given the gap between the best empirical and certified defenses across various setups (Croce & Hein, 2020; Jovanović et al., 2022; De Palma et al., 2024b), certified training schemes are commonly understood to be at odds with empirical accuracy. Expanding on the above results and complementing §5.1, section 5.2 studies whether techniques from §3, *when explicitly tuned for the purpose*, can improve empirical robustness compared to multi-step PGD on larger perturbation radii, different datasets, and shorter training schedules. Detailed comments on the relationship between the results in §5.2 and those from Gowal et al. (2018); De Bartolomeis et al. (2023); Mao et al. (2024a) are provided in appendix E.

Literature results offer partial insights into the relationship between CO and certified training schemes. First, pure IBP ($\alpha = 1$ in equation (5)), which lacks any adversarial training component, is trivially immune to CO. However, it is not applicable to all training setups (see §3.2) and typically reduces robustness compared to lower $\alpha$ values (see §2.3.2), on which we hence focus. Second, the results from De Palma et al. (2024b), which tune expressive losses for branch-and-bound certified robustness, do not display CO on benchmarks from the certified training literature. This is in spite of the fact that they mainly compute $\mathcal{L}_{\mathrm{adv}}(\boldsymbol{f}_\theta, \mathcal{B}_\epsilon(\boldsymbol{x}); y)$ via a weak one-step attack, a modified RS-FGSM with $\eta \geq 2\epsilon$, that was shown to display CO on CIFAR-10 with $\epsilon = 8/255$ on a different network and training schedule (Wong & Kolter, 2018, appendix C). As reported in De Palma et al. (2024b, appendix G.9), where performance is shown to be relatively unchanged under a more effective one-step attack (standard RS-FGSM), these results point to the lack of CO for the employed setups. Nevertheless, it is unclear from the original experiments whether $\alpha = 0$ would display CO for those experimental settings and, hence, whether expressive losses prevent it. In §5.1, we systematically investigate CO on settings popular in the relevant single-step adversarial training literature: deeper networks, different underlying attacks and datasets, shorter training schedules, and larger perturbation radii. Appendix E complements this by presenting a study on the occurrence and prevention of CO within setups from De Palma et al. (2024a). Furthermore, we show that the empirical robustness of trained models from the literature compares negatively with the results in §5.2, highlighting the importance of tuning $\alpha$ for empirical robustness.

## 5 Experimental Study

We now present the results of our experimental study on the utility of certified training schemes towards empirical robustness. Appendices B to D report omitted details and supplementary experiments.

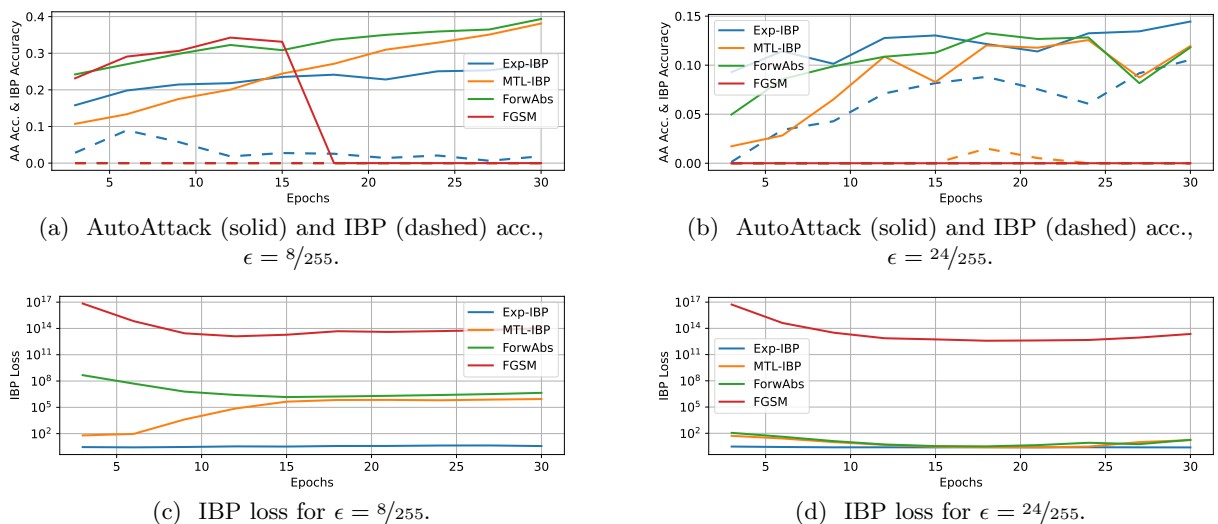

(a) AutoAttack (solid) and IBP (dashed) acc., $\epsilon = {}^8/_{255}$.

(b) AutoAttack (solid) and IBP (dashed) acc., $\epsilon = {}^{24}/_{255}$.

(c) IBP loss for $\epsilon = {}^8/_{255}$.

(d) IBP loss for $\epsilon = {}^{24}/_{255}$.

Figure 5: The use of certified training techniques on top of FGSM can prevent CO for `PreActResNet18` on the CIFAR-10 test set under perturbations of $\epsilon = {}^8/_{255}$ and $\epsilon = {}^{24}/_{255}$.

## 5.1 Preventing Catastrophic Overfitting

This section experimentally demonstrates that, on settings from the single-step adversarial training literature, Exp-IBP and ForwAbs can prevent CO when applied on top of single-step attacks.

### 5.1.1 FGSM

**Experimental setting** The seminal work from Wong et al. (2020) established that FGSM, which is arguably the most vulnerable single-step attack, suffers from CO on CIFAR-10 from as early as $\epsilon = {}^8/_{255}$. Therefore, assessing whether MTL-IBP, Exp-IBP and ForwAbs can be used to prevent CO when FGSM is used to generate the adversarial point $\boldsymbol{x}_{\mathrm{adv},\mathbb{A}}$ is a crucial qualitative test. We consider two settings: $\epsilon = {}^8/_{255}$, and the much harder $\epsilon = {}^{24}/_{255}$. In line with the single-step adversarial training literature (Andriushchenko & Flammarion, 2020; de Jorge et al., 2022; Rocamora et al., 2024) and section 3, we focus on `PreactResNet18`, trained via a cyclic learning rate schedule for 30 epochs. We run the three algorithms with a varying degree of regularization on their over-approximation tightness, plotting for each method the successful runs (i.e., preventing CO) having the smallest $\alpha$ and $\lambda$ values (see appendix C.3).

**Results** Figure 5 shows that, on both setups, all the three considered certified training schemes can prevent CO. For $\epsilon = {}^8/_{255}$, the empirical robustness of FGSM to multi-step attacks suddenly drops to near-0 roughly mid-way through training (as shown in appendix D.3, this corresponds to a sudden spike in training FGSM accuracy), and is already null at epoch 3 for $\epsilon = {}^{24}/_{255}$ (while displaying large FGSM accuracy). Differently from MTL-IBP and ForwAbs, in this setting Exp-IBP can only prevent CO while enforcing extremely low IBP loss values, resulting in non-negligible IBP accuracies on both $\epsilon = {}^8/_{255}$ and $\epsilon = {}^{24}/_{255}$. This is associated to a smaller empirical robustness than the other certified training methods on $\epsilon = {}^8/_{255}$, but not on $\epsilon = {}^{24}/_{255}$, where Exp-IBP outperforms the other algorithms also in empirical robustness. In both cases, ForwAbs attains very similar IBP losses to MTL-IBP at the end of training.

In conformity with the relevant literature, this work mainly focuses on ReLU networks. Nevertheless, in order to demonstrate their wider applicability, appendix D.8 shows that MTL-IBP, Exp-IBP, and ForwAbs can all prevent CO on a modified `PreactResNet18` that employs a SoftPlus activation function.

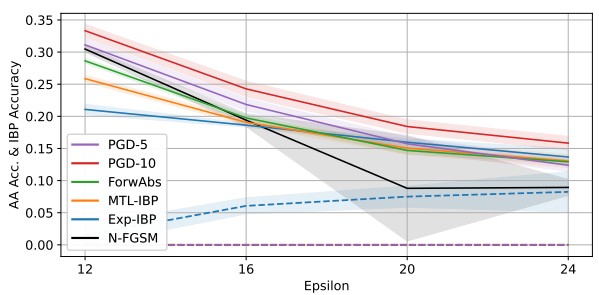 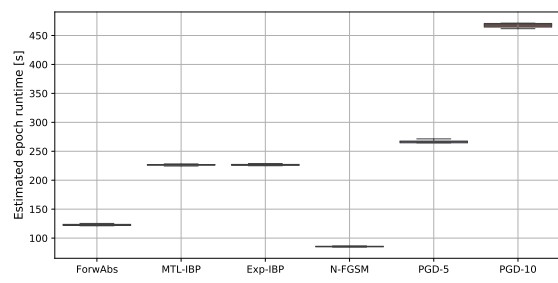

(a) AA (solid lines) and IBP (dashed) accuracies. Means over 5 repetitions and their 95% CIs.

(b) Box plots (10 runs) for the training time of a single epoch, estimated on a 80% subset of the training set.

Figure 6: Certified training techniques can prevent CO for N-FGSM when training `PreactResNet18` on CIFAR-10, overcoming the robustness of PGD-5 for $\epsilon = {}^{24}/_{255}$ while incurring less overhead.

### 5.1.2 N-FGSM

**Experimental setting**   As explained in §2.2 and §4, the addition of noise on top of FGSM can mitigate CO at lower perturbation radii without any overhead. However, previous single-step adversarial training work (Lin et al., 2023; 2024; Rocamora et al., 2024) showed that even the state-of-the-art in noise-based single-step adversarial training, N-FGSM, becomes vulnerable when $\epsilon$ is larger. In order to complement the results of figure 5, we now consider a more realistic setting where certified training schemes rely on adversarial points $\boldsymbol{x}_{\mathrm{adv},\mathbb{A}}$ generated through N-FGSM, investigating the empirical robustness cost of preventing CO at large $\epsilon$ values from the literature (Rocamora et al., 2024). As for the FGSM experiments, we use `PreactResNet18` with a 30-epoch cyclic training schedule. In these experiments we use empirical robustness to AutoAttack (AA) as the main performance criterion, and tune MTL-IBP, Exp-IBP and ForwAbs to maximize it on a holdout validation set, with $\epsilon = {}^{24}/_{255}$ for CIFAR-10 and CIFAR-100, and with $\epsilon = {}^{12}/_{255}$ for SVHN. As reported in appendix D.4, this tuning criterion will take a relatively heavy toll on standard performance. Nevertheless, §5.3 shows that, if desired, different tuning criteria may yield better trade-offs between clean and empirical robust accuracies. We then retrain on the full training set with the same chosen value for all the considered perturbation radii, reporting the test-set AA accuracy (and IBP accuracy when non-null). For this network, on CIFAR-100 and SVHN, we found the single-seed tuning employed in the rest of the paper to be a poor marker of final performance. Figure 7 reports values tuned to maximize mean validation PGD-50 accuracy on 5 seeds while preventing CO. Figure 6b shows an estimate of the training cost of each algorithm, computed on a single epoch on 80% of the CIFAR-10 training set: see appendix D.2 for further details.

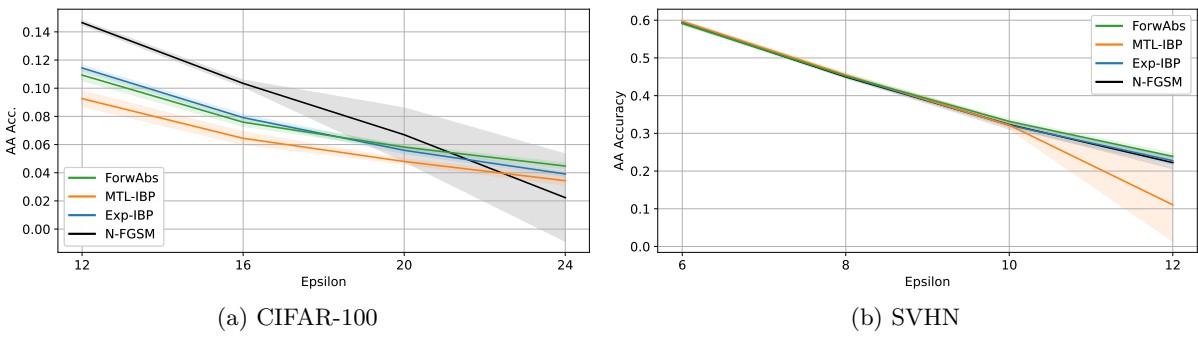

(a) CIFAR-100

(b) SVHN

Figure 7: When training `PreactResNet18`, certified training techniques can prevent CO on CIFAR-100, albeit decreasing the average empirical robustness for perturbation radii they were not tuned for. N-FGSM does not display CO for SVHN on the same network: ForwAbs results nevertheless in minor average empirical robustness improvements, while MTL-IBP induces CO at $\epsilon = {}^{12}/_{255}$ (means and 95% CIs over 5 runs).

**Results** Figure 6 shows that all the three considered algorithms can prevent CO for CIFAR-10, resulting in significantly larger robustness than N-FGSM on both $\epsilon = {}^{20}/_{255}$ and $\epsilon = {}^{24}/_{255}$. While certified training schemes pay a large price in empirical robustness for $\epsilon = {}^{12}/_{255}$, they remarkably attain stronger AA robustness than PGD-5 for $\epsilon = {}^{24}/_{255}$ with reduced runtime. Furthermore, for $\epsilon \geq {}^{20}/_{255}$ Exp-IBP yields an IBP accuracy comparable to the average AA accuracy of N-FGSM. As shown in Figure 7, CIFAR-100 and SVHN display instead worse trade-offs. While MTL-IBP, Exp-IBP and ForwAbs all prevent CO on CIFAR-100 for $\epsilon \geq {}^{20}/_{255}$, as visible from the reduced confidence intervals, they do so at a large cost in average empirical robustness for $\epsilon < {}^{24}/_{255}$. On the SVHN experiments, N-FGSM never suffers from CO. ForwAbs results in relatively small yet consistent performance improvements, while improvements from Exp-IBP are negligible. Remarkably, MTL-IBP fails to drive the IBP loss sufficiently low (cf. figure 21c) and appears to be more vulnerable to CO than N-FGSM at $\epsilon = {}^{12}/_{255}$. We ascribe this to the extreme sensitivity of MTL-IBP in the considered setup, linked to the qualitative properties of its loss on deeper networks described in §3.2 (see figure 3). We exclude MTL-IBP from the rest of the empirical study owing to this failure case. While the primary goal of this work is to study the utility of certified training schemes for empirical robustness (as opposed to outperforming the state-of-the-art in single-step adversarial training), a comparison with ELLE (Rocamora et al., 2024), a recent state-of-the-art regularizer, is presented in appendix D.7 for reference purposes. On CIFAR-10, MTL-IBP, Exp-IBP and ForwAbs all match or outperform ELLE in terms of AA accuracy for $\epsilon \geq {}^{20}/_{255}$, highlighting their particularly strong performance on easier datasets. On the other hand, on CIFAR-100 ELLE significantly outperforms certified training techniques at all perturbation radii. We refer the reader to appendix D.7 for a more detailed discussion.

In conclusion of this section, we point out that the empirical robustness costs on lower perturbation radii can be mitigated by separately tuning $\alpha$ and $\lambda$ on them (for instance, the performance of N-FGSM can be trivially preserved via $\alpha = \lambda = 0$): we refer the reader to §5.3 for results supporting this. Nevertheless, optimizing performance on lower $\epsilon$ is out of the scope of this experiment.

## 5.2 Bridging the Gap to Multi-Step Adversarial Training

We here demonstrate that the gap between certified training techniques based on single-step attacks and multi-step baselines can be reduced by training on shallower networks and with longer schedules, in some settings outperforming PGD-10 in empirical robustness without employing an adversarial training component.

### 5.2.1 Cyclic training schedule

**Experimental setting** We replicate the cyclic-schedule experiments from figures 6 and 7 on two shallower networks without skip connections. As in figures 6 and 7, $\boldsymbol{x}_{\mathrm{adv,A}}$ for certified training methods is generated via N-FGSM. First, we consider `CNN-7`, a 7-layer convolutional network from the certified training literature (Shi et al., 2021; Müller et al., 2023; Mao et al., 2023; De Palma et al., 2024b). Then, to study the impact of network depth, we also provide results on a 5-layer version, named `CNN-5` (see appendix C.3.1).

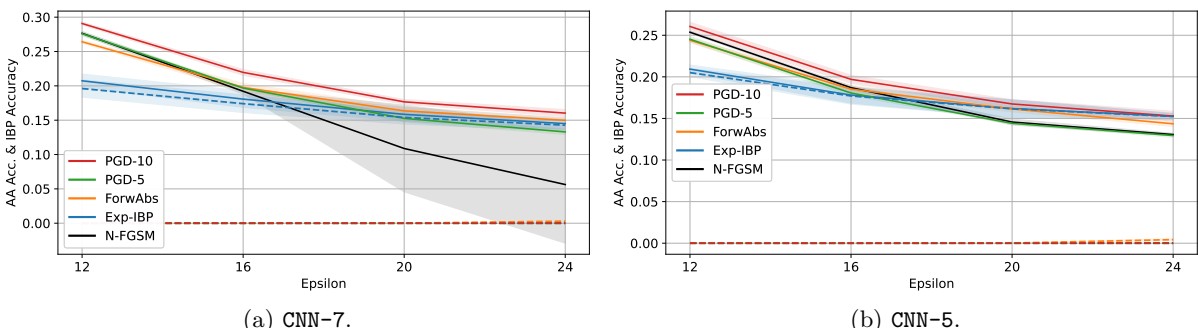

(a) `CNN-7`.                          (b) `CNN-5`.

Figure 8: When training `CNN-7` and `CNN-5` on CIFAR-10, ForwAbs and Exp-IBP prevent CO while displaying stronger empirical robustness than PGD-5 for $\epsilon \geq {}^{20}/_{255}$, and matching PGD-10 for `CNN-5` at $\epsilon = {}^{24}/_{255}$. AutoAttack (solid lines) and IBP (dashed) accuracies are reported (means and 95% CIs for 5 runs).

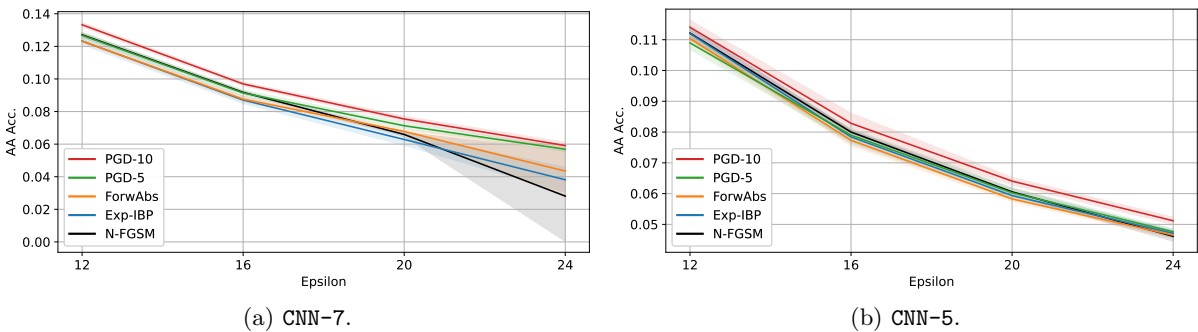

Figure 9: When training `CNN-7` and `CNN-5` for CIFAR-100, ForwAbs and Exp-IBP display better performance trade-offs than on the deeper `PreActResNet18` but still fail to improve on multi-step attacks. AutoAttack (solid lines) and IBP (dashed) accuracies are reported (means and 95% CIs for 5 runs).

We tune Exp-IBP and ForwAbs to maximize AA accuracy for $\epsilon = {}^{24}/_{255}$ on a validation set for each dataset and network, using the tuned value for all considered $\epsilon$. As discussed in §5.1 and reported in appendix D.4, this negatively impact the networks standard performance: while this is not the focus of these experiments, better robustness-accuracy trade-offs may be obtained using different tuning criteria (see §5.3).

**Results** Figure 8 demonstrates that both ForwAbs and Exp-IBP outperform PGD-5 in empirical robustness for $\epsilon \geq {}^{20}/_{255}$ on both `CNN` models, with ForwAbs preserving AA accuracy at lower epsilons, and Exp-IBP attaining significant IBP accuracy (at least as large as PGD-5's AA accuracy for $\epsilon \geq {}^{20}/_{255}$). The gap to PGD-10 is also significantly reduced compared to the `PreActResNet18` experiments, in spite of the significant CO shown by N-FGSM, the attack employed within both Exp-IBP and ForwAbs. Furthermore, as shown in appendix D.2, both Exp-IBP and ForwAbs reduce runtime compared to PGD-5 and PGD-10. Remarkably, on `CNN-5`, as shown in figure 8b, Exp-IBP matches the performance of PGD-10 for $\epsilon = {}^{24}/_{255}$, significantly outperforming N-FGSM, which does not suffer from CO in this setup. In order to prevent ForwAbs from under-performing on `CNN-5`, we found it crucial to include sufficiently low $\lambda$ values in the tuning grid: see appendix C.3.3. Finally, figure 9 shows that, while the overall performance trade-offs are greatly improved compared to `PreActResNet18`, Exp-IBP and ForwAbs fail to improve on multi-step attacks on CIFAR-100 also on the `CNN` architectures. We ascribe this to the larger number of classes, which produce a less favorable trade-off between empirical and certified robustness. Appendix D.6 shows that the `PreActResNet18` IBP loss at initialization is more than a factor $10^{10}$ larger than on the `CNN` architectures, providing an intuitive explanation for the different performance profiles.

### 5.2.2 Long training schedule

**Experimental setting** We now investigate whether certified training schemes can be more beneficial under longer training schedules. We consider a schedule of 160 epochs from the certified training literature (Shi et al., 2021) in which the learning rate is decayed twice towards the end of training, and the perturbation radius (for all methods and loss components) is gradually increased from 0 to its target value during the first 80 epochs. In order to show the benefits of tuning for each $\epsilon$, the hyper-parameters of Exp-IBP and ForwAbs ($\alpha$ and $\lambda$, respectively) are tuned to maximize validation AA accuracy individually on each setup considered in table 1, with the exception of $\epsilon = {}^{16}/_{255}$ CIFAR-10, which re-uses the values from $\epsilon = {}^{24}/_{255}$ to reduce computational overhead. As in figures 6a to 9, N-FGSM is used to generate $\boldsymbol{x}_{\text{adv},\mathbb{A}}$ for Exp-IBP and ForwAbs, except for "Exp-IBP PGD-5", which relies on PGD-5 instead. Noticing the beneficial effect of large $\alpha$ values, we also include pure IBP ($\alpha = 1$), whose loss is attack-free, in the comparison.

**Results** Table 1 shows that Exp-IBP outperforms PGD-5 on all considered CIFAR-10 setups. While empirical robustness is maximized when the IBP accuracy is large on $\epsilon = {}^{24}/_{255}$, it is not the case for $\epsilon = {}^{8}/_{255}$. Remarkably, pure IBP training outperforms all other methods for $\epsilon \geq {}^{16}/_{255}$ on CIFAR-10. Exp-IBP PGD-5 demonstrates that on CIFAR-10 certified training techniques can also improve robustness when applied on top of multi-step attacks: this appears to be useful to improve performance when single-step

Table 1: When training `CNN-7` with the long training schedule, Exp-IBP consistently improves on the average empirical robustness of PGD-5 on CIFAR-10, with IBP outperforming PGD-10 for $\epsilon \geq {}^{16}/_{255}$. Multi-step adversarial training displays the best performance on CIFAR-100. Bold entries indicate the best AA or IBP accuracy for each setting. Italics denote AA accuracy improvements on PGD-5. Means and 95% CIs for 5 runs are reported.

| | CIFAR-10 $\epsilon = {}^8/_{255}$ | | CIFAR-10 $\epsilon = {}^{16}/_{255}$ | | CIFAR-10 $\epsilon = {}^{24}/_{255}$ | | CIFAR-100 $\epsilon = {}^{24}/_{255}$ | |
|---|---|---|---|---|---|---|---|---|
| Method | AA acc. [%] | IBP acc. [%] | AA acc. [%] | IBP acc. [%] | AA acc. [%] | IBP acc. [%] | AA acc. [%] | IBP acc. [%] |
| Exp-IBP | $38.44 \pm 0.25$ | 0.00 | $20.72 \pm 1.54$ | $20.55 \pm 1.45$ | $14.79 \pm 1.61$ | $14.74 \pm 1.59$ | $3.25 \pm 0.44$ | $3.14 \pm 0.41$ |
| Exp-IBP PGD-5 | | | | | $16.45 \pm 1.32$ | $16.34 \pm 1.29$ | | |
| IBP | $29.43 \pm 0.96$ | $\mathbf{29.02 \pm 0.96}$ | $21.72 \pm 0.68$ | $\mathbf{21.62 \pm 0.68}$ | $16.73 \pm 0.43$ | $\mathbf{16.56 \pm 0.42}$ | $3.78 \pm 0.16$ | $\mathbf{3.67 \pm 0.16}$ |
| ForwAbs | $37.92 \pm 0.08$ | 0.00 | $17.61 \pm 0.38$ | 0.00 | $14.39 \pm 0.16$ | $0.46 \pm 0.17$ | $4.31 \pm 0.05$ | 0.00 |
| PGD-10 | $\mathbf{40.14 \pm 0.65}$ | 0.00 | $21.13 \pm 0.60$ | 0.00 | $15.14 \pm 0.46$ | 0.00 | $\mathbf{6.15 \pm 0.16}$ | 0.00 |
| PGD-5 | $37.32 \pm 0.71$ | 0.00 | $18.19 \pm 0.32$ | 0.00 | $11.47 \pm 0.29$ | 0.00 | $5.52 \pm 0.18$ | 0.00 |
| N-FGSM | $37.76 \pm 0.30$ | 0.00 | $7.13 \pm 11.43$ | 0.00 | $0.23 \pm 0.64$ | 0.00 | $0.02 \pm 0.06$ | 0.00 |

attacks systematically fail. While ForwAbs can significantly boost the empirical robustness of N-FGSM, it manages to outperform PGD-5 only on CIFAR-10 with $\epsilon = {}^{24}/_{255}$. Finally, the long schedule does not benefit certified training schemes on CIFAR-100, where their relative performance compared to multi-step attacks remains unvaried compared to the cyclic schedule, and for which we did not find the use of PGD-5 to generate the attacks for Exp-IBP to be beneficial.

## 5.3 Sensitivity Analysis and Performance Trade-Offs

This section presents a study of the test-set behavior of Exp-IBP and ForwAbs for varying values of their coefficients (respectively $\alpha$ and $\lambda$) when training `PreActResNet18` and `CNN-7` with the cyclic schedule. Figures 6a, 7a, 8a and 9a display results tuned to maximize AA robustness on $\epsilon = {}^{24}/_{255}$. We here focus on $\epsilon = {}^{20}/_{255}$ to potentially showcase different performance profiles, either in terms of maximizing empirical robustness or in terms of trade-offs with standard accuracy. For `PreActResNet18`, figure 10 shows that, on CIFAR-10, CO can be mitigated without incurring a significant cost in standard performance. On the other hand, maximizing empirical robustness alone, especially if with respect to AutoAttack accuracy, will result in a larger standard performance drop: this is further highlighted by the clean accuracies relative to figures 6 to 9 and table 1, which are reported in appendix D.4. Stronger regularisation is required on CIFAR-100, where low coefficient values appear to be detrimental to robustness compared to N-FGSM (corresponding to $\alpha = 0$ or $\lambda = 0$). Nevertheless, differently from the results showed in figure 7a, both Exp-IBP and ForwAbs can improve on the average empirical robustness of N-FGSM, highlighting their

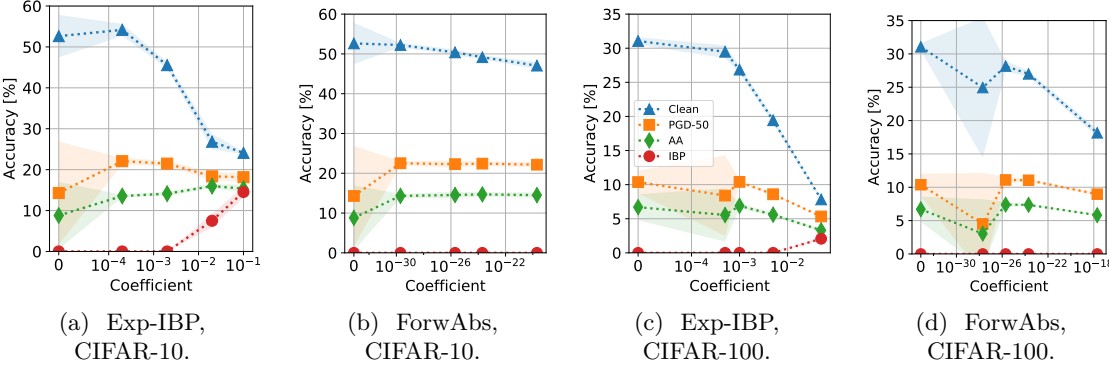

(a) Exp-IBP, CIFAR-10.     (b) ForwAbs, CIFAR-10.     (c) Exp-IBP, CIFAR-100.     (d) ForwAbs, CIFAR-100.

Figure 10: Sensitivity of Exp-IBP and ForwAbs to their respective coefficients, $\alpha$ and $\lambda$, when training `PreActResNet18` for $\ell_\infty$ perturbations of $\epsilon = {}^{20}/_{255}$. Plot 10c displays the legend for all sub-figures, which log standard accuracy (Clean), empirical robust accuracies to PGD-50 and AutoAttack (AA), and IBP verified robust accuracy on the standard test sets.

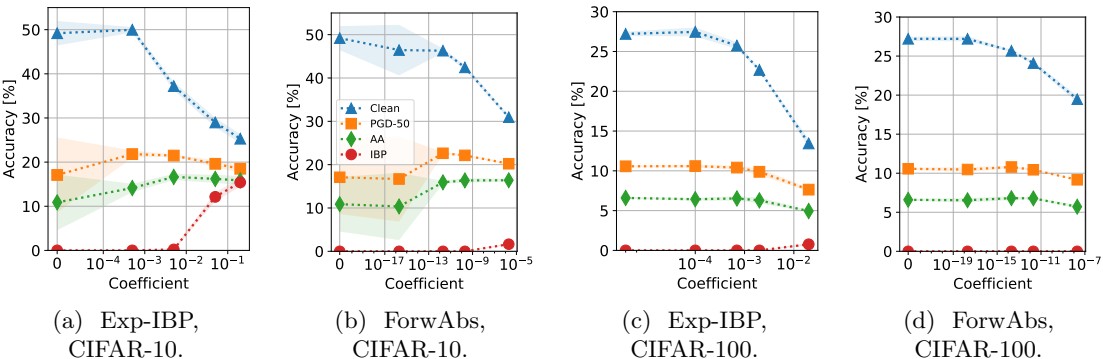

Figure 11: Sensitivity of Exp-IBP and ForwAbs to their respective coefficients, $\alpha$ and $\lambda$, on `CNN-7` for $\ell_\infty$ perturbations of $\epsilon = {}^{20}/_{255}$ using the cyclic training schedule. Plot 11b displays the legend for all sub-figures.

versatility and the potential advantages of per-instance tuning. Figure 11 shows similar trends for CIFAR-10 on `CNN-7`, where however both methods attain a larger AutoAttack accuracy, and at a smaller cost in standard performance. N-FGSM does not suffer from CO on CIFAR-100 at $\epsilon = {}^{20}/_{255}$ for `CNN-7`, where it performs competitively with PGD-5: figures 11c and 11d show that non-zero coefficients for Exp-IBP and ForwAbs are not beneficial in this setup. This is in contrast with figure 8b and table 1, suggesting that the ability of certified training schemes to enhance the empirical robustness in setups where there is no CO heavily depends on the size of the network (see appendix D.6) and on the difficulty of the learning task.

# 6 Conclusions

We presented a comprehensive empirical study on the utility of recent certified training techniques for empirical robustness, as opposed to verifiability, their original design goal. In particular, we showed that Exp-IBP can prevent catastrophic overfitting on single-step attacks for a variety of settings, outperforming some multi-step baselines in a subset of these. Furthermore, we presented a conceptually simple regularizer on the size of network over-approximations, named ForwAbs, that can achieve similar effects to Exp-IBP on top of single-step attacks while cutting down its overhead. While we believe that these results highlight the potential of certified training as an empirical defense, they also show the severe limitations of current techniques on harder datasets, ultimately calling for the development of better certified training algorithms.

### Acknowledgments

This work was partially supported by the SAIF project, funded by the "France 2030" government investment plan managed by the French National Research Agency, under the reference ANR-23-PEIA-0006; and partially funded by the French *Ministère des armées - Agence de l'innovation de défense* (Ministry of Armed Forces - Defense Innovation Agency). This publication was made possible by the use of the Factory-IA supercomputer, financially supported by the Île-de-France Regional Council. Finally, the authors are grateful to the CLEPS infrastructure from Inria Paris for providing resources and support.

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

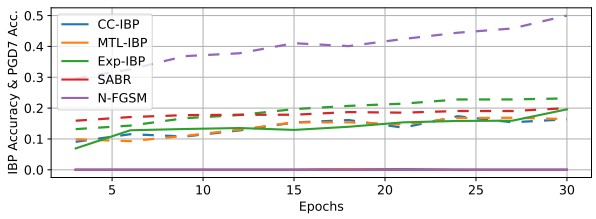
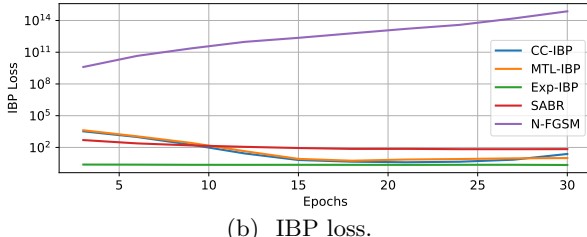

(a) Certified accuracy via IBP bounds (solid lines), empirical accuracy under PGD-7 attacks (dashed).

(b) IBP loss.

Figure 12: IBP certified robustness attained by expressive losses on the `PreActResNet18` training setup from de Jorge et al. (2022). Validation results on CIFAR-10 under perturbations of $\epsilon = {}^8/{}_{255}$.

# A  Qualitative Behavior of CC-IBP and SABR

We here present the CC-IBP and SABR losses, then expand the qualitative analysis on the behavior of expressive losses presented in §3.2. Specifically, focusing again on a `PreActResNet18` setup prevalent in the single-step adversarial training literature and studied in §5.1, we will show that they share the same qualitative behavior of MTL-IBP.

Let $\boldsymbol{x}_{\mathrm{adv},\mathbb{A}}$ be the output of the chosen attack for perturbation set $\mathcal{B}_\epsilon(\boldsymbol{x})$. SABR (Müller et al., 2023) computes the IBP loss over a subset of the perturbation set of radius $\alpha\epsilon$ centered on $\boldsymbol{x}_\alpha = \mathrm{Proj}(\boldsymbol{x}_{\mathrm{adv},\mathbb{A}}, \mathcal{B}_{\epsilon-\alpha\epsilon}(\boldsymbol{x}))$. Similarly to MTL-IBP and Exp-IBP (see §3.2), the CC-IBP loss is defined through convex combinations. Specifically, it computes the chosen loss $\mathcal{L}$ on the convex combination between the adversarial logit differences $\boldsymbol{z}_{\boldsymbol{f_\theta}}(\boldsymbol{x}_{\mathrm{adv},\mathbb{A}}, y)$ and the IBP lower bounds $\boldsymbol{l}_{\boldsymbol{f_\theta}}^{\mathcal{B}_\epsilon(\boldsymbol{x}),y}$.

**Definition A.1** *The SABR and CC-IBP losses are defined as follows:*

$$
\begin{aligned}
\mathcal{L}_\alpha^{SABR}(\boldsymbol{f_\theta}, \mathcal{B}_\epsilon(\boldsymbol{x}); y) &:= \mathcal{L}_{IBP}(\boldsymbol{f_\theta}, \mathcal{B}_{\alpha\epsilon}(\boldsymbol{x}_\alpha); y), \\
\mathcal{L}_\alpha^{CC\text{-}IBP}(\boldsymbol{f_\theta}, \mathcal{B}_\epsilon(\boldsymbol{x}); y) &:= \mathcal{L}\left( -\left[ (1-\alpha)\boldsymbol{z}_{\boldsymbol{f_\theta}}(\boldsymbol{x}_{adv,\mathbb{A}}, y) + \alpha\ \boldsymbol{l}_{\boldsymbol{f_\theta}}^{\mathcal{B}_\epsilon(\boldsymbol{x}),y} \right], y \right),
\end{aligned}
\tag{7}
$$

Figure 12 shows that, as MTL-IBP, SABR and CC-IBP fail to yield non-negligible certified accuracy via IBP. As for MTL-IBP, we gradually increase $\alpha$ and the IBP perturbation radius to improve performance (see appendix D.1). Their final IBP losses are both larger than those attained by MTL-IBP (and, transitively, by Exp-IBP). In addition, figure 13 shows that CC-IBP and SABR display the same growth trends of the MTL-IBP loss on the toy networks, further demonstrating their common similarity.

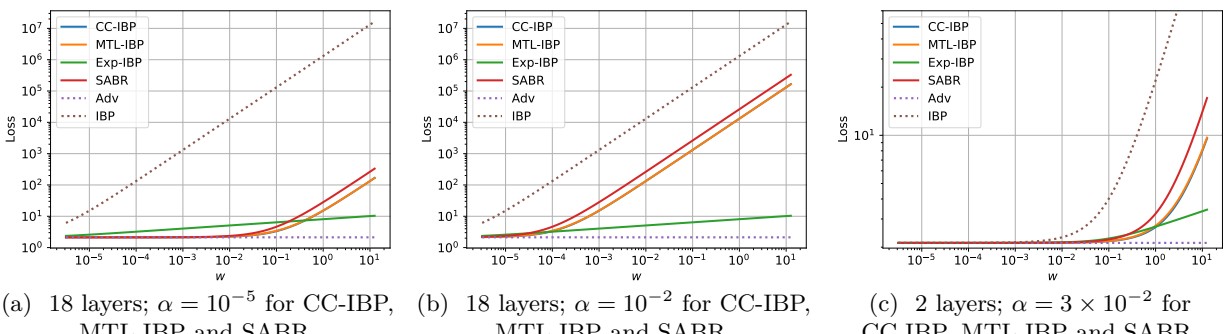

(a) 18 layers; $\alpha = 10^{-5}$ for CC-IBP, MTL-IBP and SABR.

(b) 18 layers; $\alpha = 10^{-2}$ for CC-IBP, MTL-IBP and SABR.

(c) 2 layers; $\alpha = 3 \times 10^{-2}$ for CC-IBP, MTL-IBP and SABR.

Figure 13: Sensitivity of the expressive losses on a toy network of varying depth, with $\alpha = 10^{-1}$ for Exp-IBP. For all three plots, CC-IBP displays almost identical behavior to MTL-IBP.

# B    Experimental Details

We now provide experimental details omitted from sections 3 to 5.2.

## B.1    Toy Networks

The toy neural networks employed in §3.2 are fully connected networks of depth $n$ designed to feature IBP bounds which are tunable in magnitude through a scalar parameter $w$, and explode with the depth of the network. They are defined as follows:

$$\boldsymbol{x}^1 = \text{ReLU}\left(\begin{bmatrix} w & -w \\ -w & w \end{bmatrix}\boldsymbol{x}^0\right),\ \ \boldsymbol{x}^n = \begin{bmatrix} 2 & 0 \\ 0 & 2 \end{bmatrix}\boldsymbol{x}^{n-1} + \begin{bmatrix} 3 \\ 1 \end{bmatrix},\ \ \left\{\boldsymbol{x}^k = \text{ReLU}\left(\begin{bmatrix} 2 & 0 \\ 0 & 2 \end{bmatrix}\boldsymbol{x}^{k-1}\right)\right\} \forall\ k \in [\![2, n-1]\!],$$

evaluating to:

$$\boldsymbol{x}^n = 2^{n-1}\ \text{ReLU}\left(w\begin{bmatrix} \boldsymbol{x}_0^0 - \boldsymbol{x}_1^0 \\ \boldsymbol{x}_1^0 - \boldsymbol{x}_0^0 \end{bmatrix}\right) + \begin{bmatrix} 3 \\ 1 \end{bmatrix}.$$

Let us assume that $y = 1$ (a similar reasoning holds if $y = 0$). In this context, if $w \geq 0$, the logit differences $\boldsymbol{z}_{\boldsymbol{x}^n}(\boldsymbol{x}^0, y)$ can be computed as:

$$\begin{aligned} \boldsymbol{z}_{\boldsymbol{x}^n}(\boldsymbol{x}^0, 1) &= 2^{n-1}\left[\text{ReLU}\left(w\left(\boldsymbol{x}_1^0 - \boldsymbol{x}_0^0\right)\right) - \text{ReLU}\left(w\left(\boldsymbol{x}_0^0 - \boldsymbol{x}_1^0\right)\right)\right] - 2 \\ &= w\,2^{n-1}\left[\text{ReLU}\left(\boldsymbol{x}_1^0 - \boldsymbol{x}_0^0\right) - \text{ReLU}\left(\boldsymbol{x}_0^0 - \boldsymbol{x}_1^0\right)\right] - 2 \\ &= w\,2^{n-1}\left(\boldsymbol{x}_1^0 - \boldsymbol{x}_0^0\right) - 2. \end{aligned}$$

If an $\tilde{\boldsymbol{x}}^0$ such that $\tilde{\boldsymbol{x}}_1^0 - \tilde{\boldsymbol{x}}_0^0 < 0$ belongs to the perturbation set $\mathcal{B}_\epsilon(\boldsymbol{x}^0)$, then $\boldsymbol{z}_{\boldsymbol{x}^n}(\tilde{\boldsymbol{x}}^0, 1)$ will have $(-\infty, -2]$ as image for $w \in [0, +\infty)$. Noting that the relative IBP lower bounds $\boldsymbol{l}_{\boldsymbol{x}^n}^{\mathcal{B}_\epsilon(\boldsymbol{x}^0),1}$ evaluate to $-2$ if $w = 0$ (see equation 4), that $\boldsymbol{l}_{\boldsymbol{x}^n}^{\mathcal{B}_\epsilon(\boldsymbol{x}^0),1} \leq \boldsymbol{z}_{\boldsymbol{x}^n}(\tilde{\boldsymbol{x}}^0, 1)$ for any $w \geq 0$, and that $\boldsymbol{l}_{\boldsymbol{x}^n}^{\mathcal{B}_\epsilon(\boldsymbol{x}^0),1}$ is a continuous function with respect to $w$ (as it is computed through compositions and linear combinations of continuous functions), the image of $\boldsymbol{l}_{\boldsymbol{x}^n}^{\mathcal{B}_\epsilon(\boldsymbol{x}^0),1}$ for $w \in [0, +\infty)$ will also be $(-\infty, -2]$. In addition, for any fixed $w > 0$, $\boldsymbol{l}_{\boldsymbol{x}^n}^{\mathcal{B}_\epsilon(\boldsymbol{x}^0),1}$ will decrease at least as fast as $\boldsymbol{z}_{\boldsymbol{x}^n}(\tilde{\boldsymbol{x}}^0, 1)$ (which decreases exponentially) with the depth of the network.

In order to meet the above condition, figure 3 uses $\boldsymbol{x}^0 = [-5, 5]^T$, $y = 1$ and $\mathcal{B}_{10}(\boldsymbol{x}^0)$ as perturbation. Furthermore, in order to keep the adversarial loss constant with $w$ for the purposes of figure 3, we set $\boldsymbol{x}_{\text{adv},\mathbb{A}} = [0, 0]^T$, for which $\boldsymbol{z}_{\boldsymbol{x}^n}(\boldsymbol{x}_{\text{adv},\mathbb{A}}, 1) = -2$ regardless of $w$.

## B.2    Datasets

We focus on three standard $32 \times 32$ image classification datasets: CIFAR-10 and CIFAR-100 (Krizhevsky & Hinton, 2009), and SVHN (Netzer et al., 2011). CIFAR-10 and CIFAR-100 consist of 60,000 $32 \times 32$ RGB images, with 50,000 images for training and 10,000 for testing. CIFAR-10 contains 10 classes, while CIFAR-100 contains 100 classes. SVHN consists of 73,257 $32 \times 32$ RGB images, with 73,257 images for training and 26,032 for testing.

Unless specified otherwise, for tuning purposes or when reporting validation results we use a random 20% holdout of the training set as validation set, and train on the remaining 80%. After tuning and when reporting test set results, we use the standard train and test splits for all datasets.

# C    Implementation Details and Computational Setup

This appendix details our implementation and outlines the computational setup employed to run the experiments.

## C.1    Implementation Details

Our implementation relies on PyTorch (Paszke et al., 2019) and on the public codebases from de Jorge et al. (2022); Rocamora et al. (2024); De Palma et al. (2024b). We compute IBP bounds using the `auto_LiRPA` implementation (Xu et al., 2020).

N-FGSM may return a point outside the set of allowed perturbations $\mathcal{B}_\epsilon(\boldsymbol{x})$. As a result, when using it to compute the adversarial point $\boldsymbol{x}_{\mathrm{adv},\mathbb{A}}$ for SABR, we disable the projection of the SABR perturbation subset onto the original perturbation set, simply setting $\boldsymbol{x}_\alpha = \boldsymbol{x}_{\mathrm{adv},\mathbb{A}}$ (see §3.2), which allows the implementation to meet the definition of expressivity for small $\alpha$ values.

Consistently with the single-step adversarial training literature (de Jorge et al., 2022; Rocamora et al., 2024), BatchNorm layers (Ioffe & Szegedy, 2015), which are present in all the employed models, are always kept in training mode throughout training (including during the attacks). When computing IBP bounds for expressive losses and ForwAbs terms at training time, we use the batch statistics from the current adversarial attack. For ForwAbs, these are retrieved from the `auto_LiRPA` implementation (Xu et al., 2020). The clean inputs are never fed to the network at training time, except when using FGSM to generate the attack or when training with pure IBP (in that case, the clean batch statistics are used to compute the IBP bounds). The outcome of the running statistics compute during training is systematically employed at evaluation time.

## C.2 Computational Setup

All timing measurements were carried out on an Nvidia GTX 1080Ti GPU, using 6 cores of an Intel Skylake Xeon 5118 CPU. All the other experiments were run on a single GPU each, allocated from two separate Slurm-based internal clusters. We used the following GPU models from one cluster: Nvidia V100, Nvidia RTX6000, Nvidia RTX8000, Nvidia GTX 1080Ti, Nvidia RTX2080Ti. And the following GPU models from the other cluster: Nvidia Quadro P5000, Nvidia H100.

## C.3 Training Details and Hyper-parameters

We now provide omitted training details and hyper-parameters.

### C.3.1 Network Architectures

The `PreActResNet18` and `CNN-7` architectures used in our experiments are left unvaried with respect to the implementations from de Jorge et al. (2022) and De Palma et al. (2024b), respectively.

The `CNN-5` architecture has the following structure:

1. convolutional layer with 64 $3 \times 3$ filters, stride $= 1$ and padding $= 1$, followed by BatchNorm and a ReLU activation function;

2. convolutional layer with 64 $4 \times 4$ filters, stride $= 2$ and padding $= 1$, then BatchNorm and ReLU;

3. convolutional layer with 128 $4 \times 4$ filters, stride $= 2$ and padding $= 1$, then BatchNorm and ReLU;

4. linear layer with 512 neurons, then BatchNorm and ReLU;

5. linear layer with $k$ (the number of classes) neurons.

### C.3.2 Initialization and Training Schedule

All the networks trained using MTL-IBP, Exp-IBP and ForwAbs are initialized using the specialized technique from Shi et al. (2021), which results in smaller IBP bounds at initialization. Except for the experiments from §3 and for those in table 1, where the specialized initialization is employed in order to fairly compare IBP bounds, all the networks trained via pure adversarial training and ELLE are instead initialized using PyTorch's default initialization.

All the `PreActResNet18` experiments use a short schedule popular in the literature (Andriushchenko & Flammarion, 2020; Wong et al., 2020; de Jorge et al., 2022; Rocamora et al., 2024). The batch size is set to 128, and SGD with weight decay of $5 \times 10^{-4}$ is used for the optimization. Crucially, no gradient clipping is employed, which (in addition to network depth and the lack of ramping up) we found to be a major factor behind the instability of pure IBP training (see §3.2). On CIFAR-10 and CIFAR-100 we train a PreActResnet18 for 30 epochs with a cyclic learning rate linearly increasing from 0 to 0.2 during the first

half of the training then decreasing back to 0. On SVHN the training is done for 15 epochs, with a cyclic learning rate linearly increasing from 0 to 0.05 during 6 epochs, then decreasing back to 0 for the remaining 9 epochs. Furthermore, for SVHN only, the attack perturbation radius is ramped up from 0 to $\epsilon$ during the first 5 epochs. For consistency, the `CNN` experiments from figures 8 and 9 employ the same training schedule.

All the MTL-IBP and ForwAbs experiments using the cyclic schedule gradually increase both the method coefficient (respectively $\alpha$ and $\lambda$) and the perturbation radius used to compute the IBP bounds (or their proxy $\bar{\delta}^n$) from 0 to their target value. We conduct an ablation study on MTL-IBP in the context of the experiment from figure 2 to justify this choice: we refer the reader to appendix D.1. The increase happens over the first 25 epochs for CIFAR-10 and CIFAR-100, and over the first 12 epochs for SVHN. The radius used to compute the attack is kept consistent with the adversarial training literature (that is, constant in all cases except for the first 5 epochs on SVHN). In particular, the value is increased exponentially for the first 25% of the above epochs and linearly for the rest (Shi et al., 2021), relying on the `SmoothedScheduler` from `auto_LiRPA` (Xu et al., 2020). In all cases, the attack perturbation radius is left unchanged with respect to the schedules described above for consistency with pure adversarial training.

The long training schedule used for the experiments in table 1 mirror a setup from Shi et al. (2021), widely adapted in the certified training literature (Müller et al., 2023; Mao et al., 2023; De Palma et al., 2024b). Training is carried out for 160 epochs using the Adam optimizer with a learning rate of $5 \times 10^{-4}$, decayed twice by a factor of 0.2 at epochs 120 and 140. Gradient clipping is employed, with the maximal $\ell_2$ norm of gradients equal to 10. Training starts with an epoch of clean training ("warmup"). During epochs 1 to 81, the perturbation radius (regardless of the computation it is employed for) is increased from 0 to its target value using the `SmoothedScheduler` from `auto_LiRPA` (Xu et al., 2020). For Exp-IBP and IBP, we employ a specialized regularizer (Shi et al., 2021) for the IBP bounds for the first 81 epochs (with coefficient 0.5 as done by the original authors on CIFAR-10), as commonly done in previous work (Müller et al., 2023; Mao et al., 2023; De Palma et al., 2024b).

### C.3.3 Hyper-parameters

| Experiment | Exp-IBP $\alpha$ | | MTL-IBP $\alpha$ | | ForwAbs $\tilde{\lambda}$ | |
|---|---|---|---|---|---|---|
| Figures 6a, 20a and 21a | 2 | $\times 10^{-2}$ | 1 | $\times 10^{-8}$ | 1 | $\times 10^{-24}$ |
| Figures 7a, 20g and 21b | 5 | $\times 10^{-3}$ | 1 | $\times 10^{-8}$ | 1 | $\times 10^{-18}$ |
| Figures 7b, 20c and 21c | 2 | $\times 10^{-3}$ | 2.75 | $\times 10^{-15}$ | 1 | $\times 10^{-23}$ |
| Figures 8b, 20e and 21e | 4 | $\times 10^{-1}$ | | | 1 | $\times 10^{-8}$ |
| Figures 9b, 20e and 21g | 2 | $\times 10^{-3}$ | | | 1 | $\times 10^{-10}$ |
| Figures 8a, 20d and 21d | 2 | $\times 10^{-1}$ | | | 1 | $\times 10^{-10}$ |
| Figures 9a, 20f and 21f | 2 | $\times 10^{-3}$ | | | 1 | $\times 10^{-12}$ |
| Table 1, CIFAR-10, $\epsilon = 8/255$ | 5 | $\times 10^{-3}$ | | | 1 | $\times 10^{-12}$ |
| Table 1, CIFAR-10, $\epsilon = 24/255$ and $\epsilon = 16/255$ | 7.5 | $\times 10^{-1}$ | | | 1 | $\times 10^{-8}$ |
| Table 1, CIFAR-100, $\epsilon = 24/255$ | 7.5 | $\times 10^{-1}$ | | | 1 | $\times 10^{-8}$ |

Table 2: Exp-IBP, MTL-IBP and ForwAbs coefficients for figures 6 to 9, figure 20, figure 21, and table 1.

For N-FGSM, we use the same hyper-parameters as de Jorge et al. (2022): the uniform perturbation is sampled from $[-2\epsilon, 2\epsilon]$ for every setting except for SVHN with $\epsilon = 12/255$ where the perturbation is sampled from $[-3\epsilon, 3\epsilon]$. The step size is set to $\alpha = \epsilon$ for all settings. For PGD baselines, we use a step size of $\alpha = \epsilon/4$ in all settings. Unless stated otherwise, all the certified training algorithms rely on N-FGSM to generate the adversarial attack composing their loss. To mirror the hyper-parameter in our codebase, we will report $\tilde{\lambda} = \lambda/2$ for the employed ForwAbs regularization coefficient $\lambda$.

All $\alpha$ and $\lambda$ parameters for Exp-IBP, MTL-IBP and ForwAbs Figures 6 to 9, 20 and 21 and table 1 are reported in table 2. Exp-IBP PGD5 on CIFAR-10 with $24/255$ from table 1 uses $\alpha = 0.75$. All the tuning for these experiments is done on a random validation set (see appendix B.2). On all experiments except those in figure 7, tuning is carried out by maximizing empirical robustness while above the CO threshold.

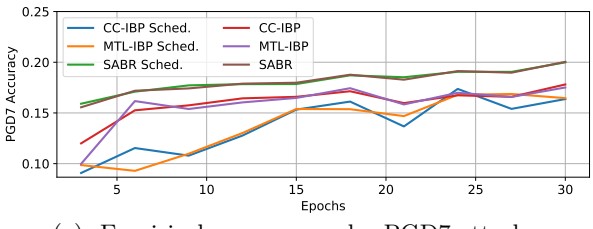
(a) Empirical accuracy under PGD7 attacks.

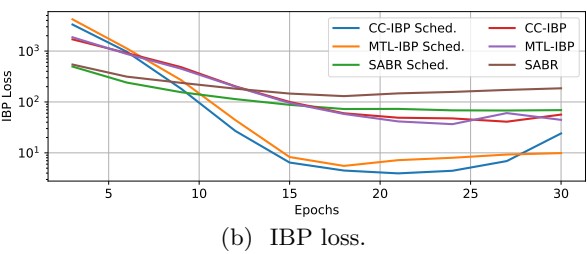
(b) IBP loss.

Figure 15: Effect of scheduling the bounding perturbation radius and $\alpha$ on the IBP certified robustness attained by CC-IBP, MTL-IBP and SABR on the `PreActResNet18` training setup from de Jorge et al. (2022). Validation results on CIFAR-10 under perturbations of $\epsilon = 8/255$.

Except where otherwise stated, the employed metric is validation AA accuracy. As explained in §5.1, certified training schemes in figure 7 were tuned using 5 seeds, maximizing PGD-50 robustness while excluding from the final hyper-parameter values any configuration displaying a strong CO behavior.

For the experiments of §3.2, §3.3 and appendix D.1, whose goal was instead to reduce the IBP loss as much as possible on the validation set, we employed the following $\alpha$ values: $\alpha = 0.1$ for Exp-IBP, $\alpha = 10^{-6}$ for CC-IBP and MTL-IBP with scheduling, $\alpha = 10^{-15}$ for CC-IBP and MTL-IBP without scheduling, $\alpha = 10^{-4}$ for SABR with scheduling and $\alpha = 10^{-9}$ for SABR without scheduling. Similarly, in figure 4, $\tilde{\lambda} = 10^{-15}$ was used for ForwAbs and $\lambda_{\ell_1} = 0.04$ for $\ell_1$-regularized N-FGSM. For all methods in this experiment, larger values among those we considered led to numerical problems or trivial behaviors, such as networks consistently outputting the same class in our implementation).

Finally, figure 5 employs the following hyper-parameters: on $\epsilon = 8/255$, $\alpha = 3 \times 10^{-2}$ for Exp-IBP, $\alpha = 10^{-8}$ for MTL-IBP, $\tilde{\lambda} = 10^{-18}$ for ForwAbs; on $\epsilon = 24/255$, $\alpha = 2.5 \times 10^{-2}$ for Exp-IBP, $\alpha = 10^{-7}$ for MTL-IBP, $\tilde{\lambda} = 2 \times 10^{-16}$ for ForwAbs. The goal of the experiment is to qualitatively discern whether certified training schemes can prevent catastrophic overfitting on FGSM. In order to determine this, we ran a series of Exp-IBP, MTL-IBP and ForwAbs experiments directly on the CIFAR-10 test set, and plotted the successful runs (in terms of preventing CO) with the smallest $\alpha$ or $\tilde{\lambda}$ values (among those tried).

In conclusion of this appendix, we show a preliminary version of figure 8b resulting from insufficient tuning of the ForwAbs coefficient $\lambda$. Specifically, when tuning for the experiment, we first focused on excessively large coefficients, selecting $\tilde{\lambda} = 10$ as the best-performing setup in terms of validation AA accuracy (as opposed to $\tilde{\lambda} = 10^{-8}$ used for figure 8b). Figure 14 shows the result: while ForwAbs achieves large IBP accuracy on `CNN-5`, further confirming its utility towards certified robustness, it attains low empirical robustness throughout the experiment. Noticing instead the strong empirical robustness of ForwAbs with $\tilde{\lambda} = 10^{-10}$ on `CNN-7` (figure 8a), we extended the tuning on `CNN-5` to smaller $\tilde{\lambda}$ values. We then found $\tilde{\lambda} = 10^{-8}$ to yield stronger validation AA accuracy than $\tilde{\lambda} = 10$, hence producing figure 8b.

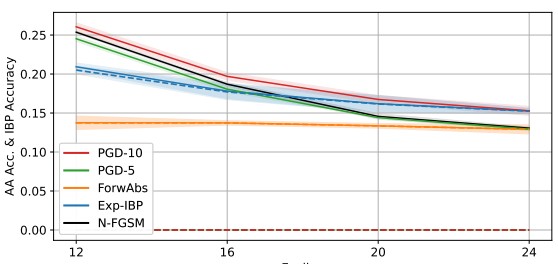

Figure 14: Results of insufficient tuning of ForwAbs for `CNN-5` on CIFAR-10, setup of figure 8b.

# D  Additional Experimental Results

This appendix displays supplementary experimental results.

### D.1    Supplement to Section 3.2, Scheduling Ablation

We here present experimental results omitted from §3.2 and appendix A. Figure 16 displays the behavior of the training IBP loss for Exp-IBP. In particular, it shows that the training loss drops from more than $10^{17}$ to 2.67 from the first to the second epoch, demonstrating the large degree of variability in IBP loss for the employed `PreActResNet18` training setup. In this context, as shown in figure 3, Exp-IBP displays less sensitivity to the magnitude of the IBP loss, proving more effective for the purposes of decreasing the IBP loss. In fact, the other losses from equation (5) display numerical problems that prevent successful training for large $\alpha$ coefficients (and for pure IBP, which corresponds to $\alpha = 1$ for any expressive loss), leading to aborted runs in our `PyTorch` implementation.

Figure 15 studies the effect of scheduling (gradually increasing from 0 to their target value in 25 epochs) both $\alpha$ and the perturbation radius $\epsilon$ employed to compute the IBP lower bounds $\boldsymbol{l}_{\boldsymbol{f_\theta}}^{\mathcal{B}_\epsilon(\boldsymbol{x}),y}$. Excluding trivial outcomes, such as networks systematically outputting the same class, for CC-IBP, MTL-IBP and SABR, we were unable to reach validation IBP loss values below 44 without scheduling. The scheduling allows the use of larger $\alpha$ values, resulting in lower IBP loss value for CC-IBP, MTL-IBP and SABR: respectively from 56.47 to 24.06, from 44.50 to 9.89, and from 185.51 to 69.09. These improvements come with a slight de-

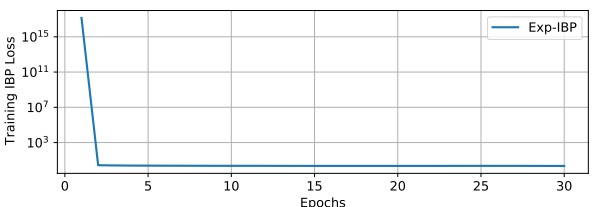

Figure 16:  Training IBP loss of the Exp-IBP experiment from figure 2.

crease in robustness to PGD-7 (a PGD attack with 7 steps). Nevertheless, CC-IBP, MTL-IBP and SABR are unable to attain non-negligible certified accuracy via IBP in this context. In view of these results, we employ the same scheduling for ForwAbs (see appendix C.3.2).

### D.2    Training Overhead

Figure 6b shows the per-epoch training overhead of MTL-IBP, Exp-IBP, ForwAbs, PGD-5 and PGD-10 with respect to N-FGSM (which is used to compute the attack for MTL-IBP, Exp-IBP, ForwAbs) when training a `PreActResNet18` on 80% of the CIFAR-10 training set using the cyclic schedule, complementing the information provided in figure 4. As described in §3.3, ForwAbs has minimal overhead with respect to N-FGSM. On the other hand, MTL-IBP and Exp-IBP display a runtime slightly smaller than PGD-5. As expected, PGD-10 training requires almost twice the time as PGD-5, and more than half the time as MTL-IBP and Exp-IBP. Figure 17, instead, plots the respective training overheads when training `CNN-7` and `CNN-5` using the cyclic schedule. The relative runtimes across methods remain similar to `PreActResNet18`, with the runtime of each algorithm markedly smaller on the `CNN` models, and for the 5-layer network. Differently from the experiments in §5, all these runtime measurements were carried out on the same machine under constant load (see also appendix C.2).

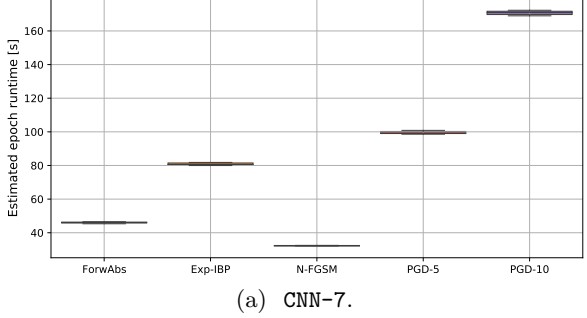

(a) `CNN-7`.

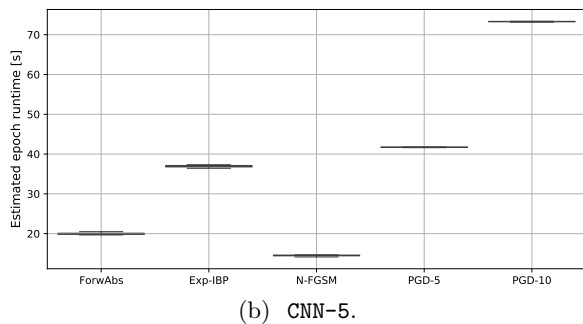

(b) `CNN-5`.

Figure 17:  Box plots (10 repetitions) for the CIFAR-10 training time of a single epoch (using a 80% subset of the training set) on `CNN-7` and `CNN-5`.

Figure 18 explicitly plots trade-offs between runtime and empirical robustness. In particular, figures 18a and 18c respectively plot the runtimes from figures 17a and 6b against the AA accuracies reported for CIFAR-10 at $\epsilon = {}^{24}/_{255}$ within figures 6a and 8a. Differently from the cyclic schedule, the long schedule (160 epochs) used for the experiments of table 1 does not have a constant training cost throughout the epochs (see appendix C.3.2): all methods start with a warmup epoch that requires additionally evaluating the network on clean inputs, and methods using IBP bounds (Exp-IBP and IBP) employ the regularizer from Shi et al. (2021) to control the bounds in earlier epochs, increasing their overhead until epoch 81. Figure 18b plots the AA accuracies for CIFAR-10 at $\epsilon = {}^{24}/_{255}$ from table 1 against the per-epoch runtimes from figure 17a for N-FGSM, PGD-5, PGD-10 and ForwAbs (hence providing an estimate of long-schedule runtime after warmup, on 80% of the CIFAR-10 training set), and against the per-epoch runtimes (also computed on 80% of the training set, after warmup) of Exp-IBP and IBP when the bounds regularizer is active. Across the three setups, both ForwAbs and Exp-IBP consistently outperform PGD-5, with the former incurring a relatively small overhead compared to N-FGSM. On the long schedule, IBP outperforms all the other algorithms despite its low runtime, including the stronger multi-step baseline PGD-10.

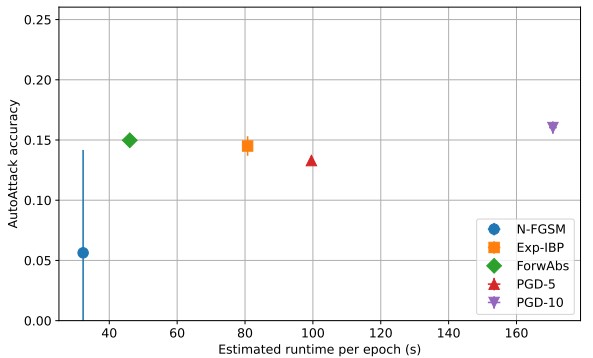

(a) `CNN-7`, CIFAR-10, $\epsilon = {}^{24}/_{255}$, setup of figure 8a.

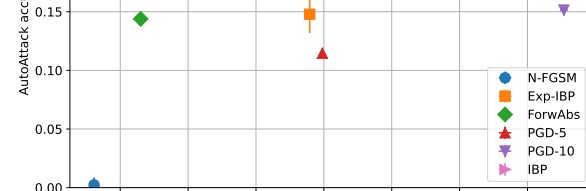

(b) `CNN-7`, CIFAR-10, $\epsilon = {}^{24}/_{255}$, setup of table 1

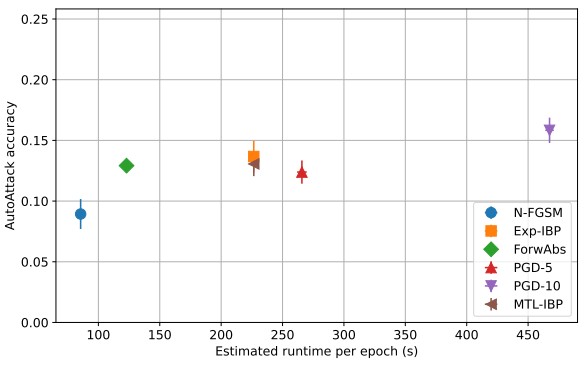

(c) `PreactResNet18`, CIFAR-10, $\epsilon = {}^{24}/_{255}$, setup of figure 6a

Figure 18: Trade-offs between estimated per-epoch runtime and AA accuracy on CIFAR-10 for $\epsilon = {}^{24}/_{255}$.

### D.3 Supplement to FGSM experiments

Figure 19 plots the training adversarial accuracy computed via FGSM, the attack employed to generate $\boldsymbol{x}_{\mathrm{adv},\mathbb{A}}$ for all training methods in this experiment, associated to the plots in figure 5. The behavior of the training adversarial accuracy for FGSM, which spikes up for $\epsilon = {}^{8}/_{255}$ roughly when its validation PGD-7 accuracy goes towards 0 (cf. figure 5), and attains very large values for $\epsilon = {}^{24}/_{255}$ in spite of its null validation PGD-7 accuracy, is a clear marker of catastrophic overfitting.

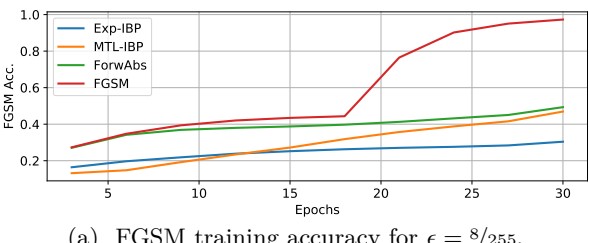 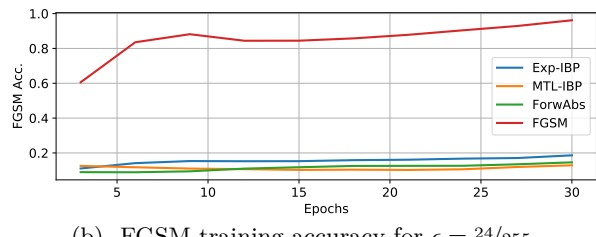

(a)  FGSM training accuracy for $\epsilon = {}^8\!/255$.  (b)  FGSM training accuracy for $\epsilon = {}^{24}\!/255$.

Figure 19:  Training FGSM accuracy for the experiments from figure 5.

### D.4   Clean Accuracies and IBP Loss for N-FGSM Experiments

We here provide omitted clean accuracies and IBP losses for the experiments for figures 6 to 9 and table 1. Figure 20 provides the results for the cyclic training schedule on `PreActResNet18`, `CNN-7` and `CNN-5`. It is clear that the positive effects associated to the certified training schemes (the mitigation or prevention of catastrophic overfitting in §5.1 and the empirical robustness improvements detailed in §5.2) come at a cost in clean accuracy, which is a limitation of these methods and of enforcing low IBP loss. Generally speaking, as visible in figure 21, Exp-IBP attains significantly better trade-offs between clean accuracy and IBP loss, especially when compared to MTL-IBP. As a result, Exp-IBP is to be preferred if IBP accuracy is the primary metric. Table 3 presents the clean accuracies and IBP losses pertaining to the long schedule experiment from table 1, additionally presenting a comparison between the IBP losses of the certified training schemes tuned for empirical robustness and the adversarial training baselines whose IBP loss was shown in figure 1. In all cases, certified training schemes decrease the IBP loss compared to the adversarial training baselines at a cost in standard accuracy, with a larger cost associated to a greater reduction in IBP loss. We conclude by pointing out that an alternative tuning criterion (considering trade-offs between standard accuracy and empirical robustness) could result in more favorable trade-offs. This was not the focus of the presented experiments.

Table 3:  Clean accuracies and IBP losses for the experiments in table 1.

| Dataset | $\epsilon$ | Method | Clean acc. [%] | IBP loss |
|---|---|---|---|---|
| CIFAR-10 | $\frac{8}{255}$ | Exp-IBP | $76.30 \pm 0.43$ | $(4.90 \pm 0.55) \times 10^4$ |
| | | IBP | $39.37 \pm 1.51$ | $1.92 \pm 0.01$ |
| | | FORWABS | $74.26 \pm 0.28$ | $(2.05 \pm 0.11) \times 10^4$ |
| | | PGD-10 | $77.07 \pm 0.31$ | $(9.79 \pm 0.85) \times 10^5$ |
| | | PGD-5 | $79.81 \pm 0.38$ | $(1.72 \pm 0.11) \times 10^6$ |
| | | N-FGSM | $77.28 \pm 0.26$ | $(1.91 \pm 0.14) \times 10^6$ |
| | $\frac{16}{255}$ | Exp-IBP | $31.30 \pm 2.13$ | $2.15 \pm 0.02$ |
| | | IBP | $31.65 \pm 0.55$ | $2.13$ |
| | | FORWABS | $46.54 \pm 0.31$ | $(7.72 \pm 0.37) \times 10^1$ |
| | | PGD-10 | $58.44 \pm 0.24$ | $(3.57 \pm 0.33) \times 10^5$ |
| | | PGD-5 | $64.07 \pm 0.14$ | $(1.01 \pm 0.13) \times 10^6$ |
| | | N-FGSM | $62.38 \pm 9.16$ | $(2.93 \pm 2.66) \times 10^6$ |
| | $\frac{24}{255}$ | Exp-IBP | $23.03 \pm 4.97$ | $2.25 \pm 0.02$ |
| | | Exp-IBP PGD-5 | $27.32 \pm 2.94$ | $2.23 \pm 0.02$ |
| | | IBP | $23.51 \pm 2.63$ | $2.22 \pm 0.01$ |
| | | FORWABS | $30.15 \pm 0.37$ | $8.73 \pm 0.53$ |
| | | PGD-10 | $39.99 \pm 0.60$ | $(2.60 \pm 0.42) \times 10^4$ |
| | | PGD-5 | $49.31 \pm 0.37$ | $(2.13 \pm 0.22) \times 10^5$ |
| | | N-FGSM | $57.46 \pm 9.60$ | $(3.31 \pm 1.55) \times 10^6$ |
| CIFAR-100 | $\frac{24}{255}$ | Exp-IBP | $6.85 \pm 1.29$ | $4.49 \pm 0.02$ |
| | | IBP | $7.29 \pm 0.61$ | $4.45 \pm 0.01$ |
| | | FORWABS | $16.85 \pm 0.67$ | $(5.79 \pm 0.36) \times 10^1$ |
| | | PGD-10 | $23.56 \pm 0.61$ | $(2.19 \pm 0.19) \times 10^5$ |
| | | PGD-5 | $28.53 \pm 0.46$ | $(6.12 \pm 0.32) \times 10^5$ |
| | | N-FGSM | $29.07 \pm 3.01$ | $(6.50 \pm 1.92) \times 10^6$ |

Figure 20: Clean accuracies for the experiments from figures 6, 7, 8 and 9. Means and 95% confidence intervals over 5 repetitions.

(a) `PreActResNet18`, CIFAR-10.

(b) `PreActResNet18`, CIFAR-100.

(c) `PreActResNet18`, SVHN.

(d) `CNN-7`, CIFAR-10.

(e) `CNN-5`, CIFAR-10.

(f) `CNN-7`, CIFAR-100.

(g) `CNN-5`, CIFAR-100.

Figure 21: IBP losses of methods from §3 for the experiments from figures 6, 7, 8 and 9. Means and standard deviations over 5 repetitions.

(a) `PreActResNet18`, CIFAR-10.

(b) `PreActResNet18`, CIFAR-100.

(c) `PreActResNet18`, SVHN.

(d) `CNN-7`, CIFAR-10.

(e) `CNN-5`, CIFAR-10.

(f) `CNN-7`, CIFAR-100.

(g) `CNN-5`, CIFAR-100.

Table 4: Validation empirical robustness of the last model and of the model corresponding to the best PGD-50 accuracy when training `PreActResNet18` with the cyclic schedule (means and 95% CIs for 5 runs).

| Dataset | $\epsilon$ | Method | Final AA acc. [%] | Selected AA acc. [%] |
|---|---|---|---|---|
| CIFAR-10 | $\frac{20}{255}$ | Exp-IBP | $15.88 \pm 1.39$ | $15.42 \pm 2.37$ |
| | | ForwAbs | $14.22 \pm 1.68$ | $14.96 \pm 1.29$ |
| | | N-FGSM | $10.34 \pm 5.43$ | $12.80 \pm 2.34$ |
| CIFAR-100 | $\frac{24}{255}$ | Exp-IBP | $3.74 \pm 1.01$ | $3.92 \pm 0.96$ |
| | | ForwAbs | $4.14 \pm 0.80$ | $4.08 \pm 0.78$ |
| | | N-FGSM | $2.20 \pm 2.29$ | $4.34 \pm 0.50$ |

## D.5 Validation CO Study

This appendix presents a study on the behavior of Exp-IBP and ForwAbs over the course of training epochs, aimed at providing further insights into their effect on CO when applied on top of N-FGSM. Specifically, we evaluate on a holdout set of 1000 images, and train on the remainder of the original training set. We compute the PGD-50 accuracy at each epoch, and store two models for a more accurate empirical robustness evaluation via AutoAttack: the one corresponding to the best PGD-50 accuracy (we call this the selected model), and the final model. We focus on two `PreActResNet18` setups: CIFAR-10 at $\epsilon = {}^{20}/{}_{255}$, and CIFAR-100 at $\epsilon = {}^{24}/{}_{255}$, which are the benchmarks in which N-FGSM displays the worst average AutoAttack accuracy within figures 6 and 7a.

Table 4 shows that, on both settings, Exp-IBP and ForwAbs significantly reduce the variability between the final and the selected models compared to N-FGSM, further confirming their ability to prevent CO. On CIFAR-10, both methods attain stronger empirical robustness than the selected N-FGSM models, showing that they are preferable to early stopping with an expensive attack on this dataset. While this is not immediately confirmed on the harder CIFAR-100, we point out that a tuning process aware of early stopping may likely bring to a different conclusion: this is beyond the scope of the experiment. Exp-IBP and ForwAbs were run with the following coefficients: on CIFAR-10, $\alpha = 2 \times 10^{-2}$ for Exp-IBP, and $\tilde{\lambda} = 10^{-22}$ for ForwAbs; on CIFAR-100, $\alpha = 5 \times 10^{-3}$ for Exp-IBP, and $\tilde{\lambda} = 10^{-18}$ for ForwAbs. These coincide with the coefficients employed for figures 6 and 7a, except for ForwAbs on CIFAR-10, where the new runs for this experiment showed some signs of CO for these dataset splits using $\tilde{\lambda} = 10^{-24}$ and hence prompted us to employ a larger coefficient for this appendix, pointing to some degree of experimental variability (cf. figure 10b, which does not display CO even for significantly smaller coefficient values). We remark that the ForwAbs coefficient for figure 6 was tuned using a single seed, on a different dataset split, at $\epsilon = {}^{24}/{}_{255}$: variability could likely be reduced by tuning on multiple seeds and for the same perturbation radius.

## D.6 Effect of Model Architecture

The experimental results in §5 show that the relative efficacy of certified training schemes towards empirical robustness improves on shallower networks. The present appendix is devoted to providing further insights on this phenomenon.

### D.6.1 Effect of Model Architecture on Method Performance

Table 5 presents an analysis of the effect of model size on the performance of both multi-step adversarial training and certified training techniques, grouping in table form the results from figures 6a, 8a, and 8b on

Table 5: Effect of model architecture on AutoAttack accuracy on CIFAR-10 with $\epsilon = {}^{24}/{}_{255}$. Results from figures 6a, 8a, and 8b (means and standard deviations for 5 runs).

| Architecture | Exp-IBP AA acc. [%] | ForwAbs AA acc. [%] | PGD-5 AA acc. [%] | PGD-10 AA acc. [%] |
|---|---|---|---|---|
| `PreActResNet18` | $13.67 \pm 1.30$ | $12.92 \pm 0.34$ | $12.39 \pm 0.96$ | $15.83 \pm 1.05$ |
| `CNN-7` | $14.50 \pm 0.81$ | $14.98 \pm 0.37$ | $13.29 \pm 0.37$ | $16.03 \pm 0.52$ |
| `CNN-5` | $15.26 \pm 0.35$ | $14.36 \pm 0.32$ | $12.94 \pm 0.23$ | $15.28 \pm 0.58$ |

Table 6: IBP loss at initialization for the network architectures considered in this work, computed on the CIFAR-10 training set against perturbations of radius $\epsilon = {}^{24}/{}_{255}$ (means and standard deviations for 5 runs).

| Architecture | IBP loss |
|---|---|
| PreActResNet18 | $(2.20 \pm 0.06) \times 10^{16}$ |
| CNN-7 | $(8.67 \pm 0.18) \times 10^{5}$ |
| CNN-5 | $(1.11 \pm 0.01) \times 10^{4}$ |

CIFAR-10 with $\epsilon = {}^{24}/{}_{255}$. Differently from §5, where the focus was on relative performance across methods, here we concentrate on the impact of model architecture on the performance of each method. All considered algorithms attain larger empirical robustness on CNN-7 compared to PreActResNet18. Remarkably, and differently from all the other techniques, the AutoAttack accuracy of Exp-IBP is maximized on CNN-5, suggesting that smaller networks may be particularly beneficial to expressive losses under shorter training schedules.

### D.6.2 IBP Losses at Initialization across Architectures

In order to provide further insights on the results from §5 and table 5, we compute the IBP loss at initialization on the three network architectures employed in this work, which provides an indication of the size of the IBP network over-approximation. Specifically, we measure the average IBP loss over the standard CIFAR-10 training set, against perturbations of radius $\epsilon = {}^{24}/{}_{255}$. Table 6 shows that, as expected, the network bounds explode with the model depth, with PreActResNet18 displaying an IBP loss at initialization that is several orders of magnitude larger than those of the two employed CNN architectures. As a result, the significantly smaller IBP loss values associated with maximal AutoAttack accuracy (see figure 21) come at a larger cost in terms of empirical robustness, resulting in worse performance trade-offs. Finally, CNN-7 features an IBP loss almost two orders of magnitude larger than CNN-5, explaining the qualitative differences in behavior from figure 8.

### D.7 Comparison with ELLE

The main objective of this work is to demonstrate that certified training techniques can be successfully employed towards empirical robustness, hence beyond their original design goal. As part of the provided evidence, section 5.1 shows that Exp-IBP and ForwAbs can prevent CO on settings common to the single-step adversarial training literature. In order to contextualize their performance for the task, this appendix provides a comparison of the performance of Exp-IBP, MTL-IBP and ForwAbs with ELLE-A (Rocamora et al., 2024), a state-of-the-art method designed to prevent catastrophic overfitting.

The experimental setup is the one from figures 6 and 7 (a PreActResNet18 trained with a cyclic training schedule). As for the certified training techniques, we use N-FGSM as underlying adversarial attack. The runtime overhead of ELLE-A, which is slightly smaller than MTL-IBP and Exp-IBP, and significantly larger than ForwAbs on an Nvidia GTX 1080Ti GPU, is reported in figure 22d. While the original work (Rocamora et al., 2024) reports less overhead for ELLE-A, which requires 3 batched forward passes to compute its regularization term, we found its overhead to be heavily dependent on the GPU model, with newer models leading to a faster execution of the batched forward pass (the original work relies on an Nvidia A100 SXM4 GPU). We tuned the ELLE-A hyper-parameter $\lambda_{\text{ELLE}}$, which controls the amount of regularization imposed on a notion of local linearity proposed by the authors, consistently with the way MTL-IBP, Exp-IBP and ForwAbs were tuned for figures 6 and 7. The tuning resulted in the following values: $\lambda_{\text{ELLE}} = 6000$ for CIFAR-10 (which maximizes both the validation PGD-50 and AA accuracy), $\lambda_{\text{ELLE}} = 3000$ for CIFAR-100, and $\lambda_{\text{ELLE}} = 1000$ for SVHN.

As reported in Figure 22, ELLE-A prevents CO across all settings. Remarkably, on CIFAR-10, certified training techniques match or outperform ELLE-A in AA accuracy for $\epsilon \geq {}^{20}/{}_{255}$, with Exp-IBP outperforming ELLE-A while also producing non-negligible certified robustness via IBP. We believe this suggests that verifiability and empirical robustness are not conflicting objectives in this setup. Furthermore, ForwAbs performs at least competitively with ELLE-A on all the considered epsilons while reducing its overhead. The situation is drastically different on the harder CIFAR-100, where ELLE-A markedly outperforms certified training schemes. We speculate the employed network may lack the capacity to sustain tight over-approximations

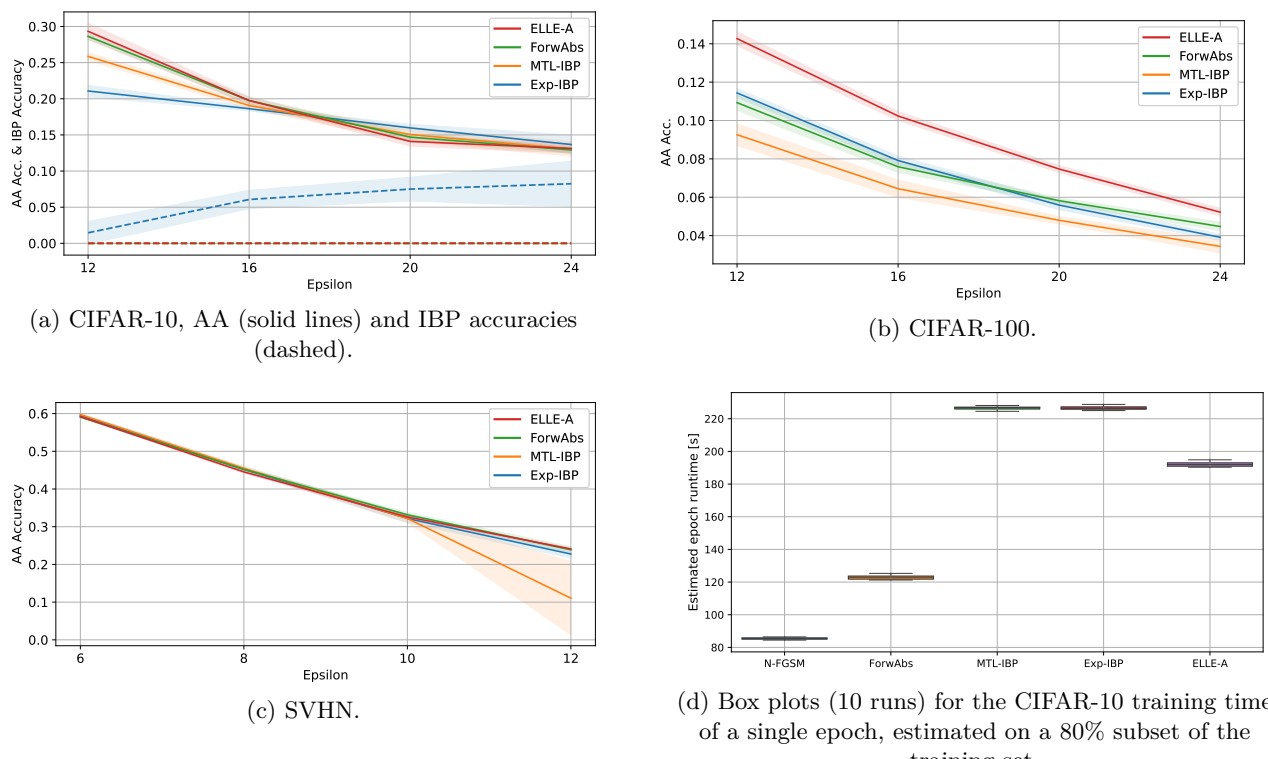

(a) CIFAR-10, AA (solid lines) and IBP accuracies (dashed).

(b) CIFAR-100.

(c) SVHN.

(d) Box plots (10 runs) for the CIFAR-10 training time of a single epoch, estimated on a 80% subset of the training set.

Figure 22: Comparison of certified training techniques with ELLE-A (Rocamora et al., 2024), a the state-of-the-art regularizer for singe-step adversarial training. Setup from figures 6 and 7. We report means over 5 repetitions and their 95% confidence intervals.

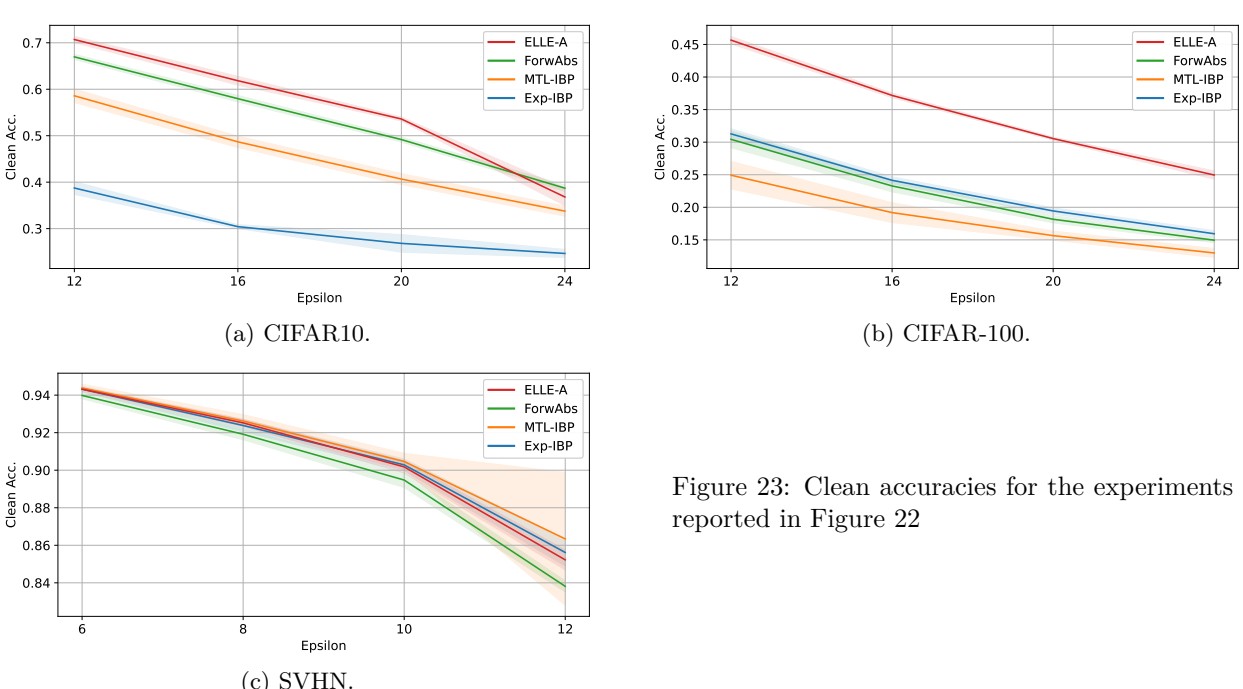

(a) CIFAR10.

(b) CIFAR-100.

(c) SVHN.

Figure 23: Clean accuracies for the experiments reported in Figure 22

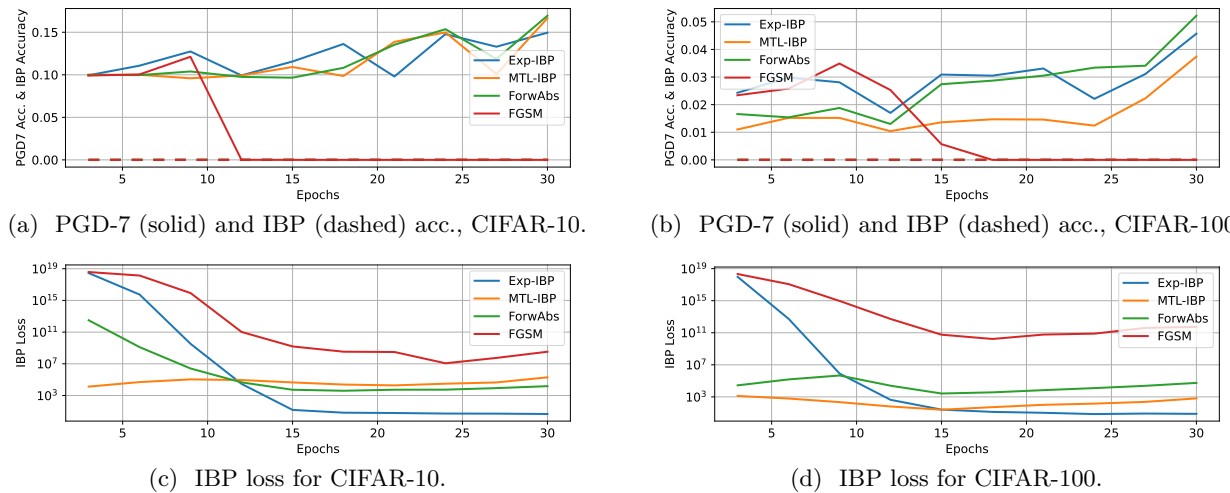

(a)  PGD-7 (solid) and IBP (dashed) acc., CIFAR-10.  (b)  PGD-7 (solid) and IBP (dashed) acc., CIFAR-100.

(c)  IBP loss for CIFAR-10.  (d)  IBP loss for CIFAR-100.

Figure 24:   When applied on top of FGSM, certified training techniques can prevent CO beyond ReLU networks.  Results with a modified `PreActResNet18` employing SoftPlus activations, for perturbations of $\epsilon = {}^{24}/_{255}$ on the CIFAR-10 and CIFAR-100 test sets.

while preserving empirical robustness. On SVHN, except MTL-IBP which fails owing to the sensitivity of its loss as discussed in §5.1, all the algorithms attain similar performance profiles, with the most robust algorithm depending on the perturbation radius. Figure 23 reports the corresponding clean accuracies: on average ELLE-A has less impact on clean accuracy than certified training schemes, except for SVHN, where expressive losses display larger standard performance, and on CIFAR-10 for $\epsilon = {}^{24}/_{255}$, where ForwAbs does. In summary of this appendix, we argue that, on easier datasets such as CIFAR-10, certified training techniques such as Exp-IBP and ForwAbs should be considered as strong baselines for CO prevention by the single-step adversarial training community.

### D.8   Preventing Catastrophic Overfitting on SoftPlus

We here present results demonstrating that certified training schemes can enhance empirical robustness beyond piecewise-linear networks. In particular, we focus on a modified `PreActResNet18` architecture, for which each ReLU is replaced by a SoftPlus activation, relying on the training schedule from figure 5. Owing to the monotonicity of SoftPlus, IBP bounds can be computed using the procedure outlined in §2.3.1 and using the `auto_LiRPA` library, which we also employ for ReLU networks (see appendix C.1). We leave the ForwAbs implementation unvaried with respect to §3.3.

Figure 24 provides results for CIFAR-10 and CIFAR-100 at $\epsilon = {}^{24}/_{255}$, for which we show FGSM to be highly vulnerable to CO. Mirroring the experimental procedure for figure 5, we demonstrate that MTL-IBP, Exp-IBP and ForwAbs can all prevent CO on this setting. Similarly to the ReLU experiments, this involves significantly decreasing the IBP loss compared to FGSM. Figure 24 plots the following coefficient values: on CIFAR-10, $\alpha = 5 \times 10^{-3}$ for Exp-IBP, $\alpha = 2 \times 10^{-11}$ for MTL-IBP, $\tilde{\lambda} = 10^{-20}$ for ForwAbs; on CIFAR-100, $\alpha = 1 \times 10^{-2}$ for Exp-IBP, $\alpha = 10^{-10}$ for MTL-IBP, $\tilde{\lambda} = 10^{-20}$ for ForwAbs.

## E   Previous Work on the Empirical Accuracy of Certified Training

We now comment on the relationship between §5 and literature results, explicit or implicit, on the empirical accuracy of certified training algorithms (see §4). In particular, §E.2 provides additional experiments studying CO within setups from the certified training literature (De Palma et al., 2024b).

### E.1 Empirical Robustness Evaluations

While this was not the focus of their works, Gowal et al. (2018); Mao et al. (2024a) both demonstrated that certified training schemes (including IBP for Gowal et al. (2018), IBP, SABR and MTL-IBP for Mao et al. (2024a)) outperform multi-step adversarial training on large perturbation radii on MNIST for relatively shallow convolutional networks. Specifically, Gowal et al. (2018) showed that IBP outperforms PGD-7 training in terms of PGD-200-10 (PGD with 200 iterations and 10 restarts) accuracy for $\epsilon \in \{0.3, 0.4\}$, when using a 100-epoch training schedule. Mao et al. (2024a) tuned the certified training schemes to maximize certified robustness, and then positively compared their empirical robustness, measured through the attacks within MN-BaB, a complete verifier based on branch-and-bound (Ferrari et al., 2022), to the AA accuracy of PGD-5-3 training for $\epsilon = 0.3$ (using a 70-epoch training schedule). Moreover, De Bartolomeis et al. (2023) report that, on MNIST, IBP (Shi et al., 2021) outperforms adversarial training based on a 10-step AutoPGD (Croce & Hein, 2020) attack in terms of AA accuracy for $\epsilon \in \{0.1, 0.2, 0.3, 0.4\}$. However, they employ inconsistent training schedules and optimizers across methods, and their adversarial training baseline severely under-performs compared to other works (Mao et al., 2024a). We omit MNIST from our main study owing to its relative simplicity, and to our focus on settings typically considered in the single-step adversarial training literature (de Jorge et al., 2022; Rocamora et al., 2024; Andriushchenko & Flammarion, 2020).

Furthermore, Gowal et al. (2018); Mao et al. (2024a) reported that certified training schemes (IBP for Gowal et al. (2018), SABR and MTL-IBP for Mao et al. (2024a)) can outperform multi-step attacks in terms of empirical robustness using `CNN-7` for $\epsilon = 8/255$ on CIFAR-10. Gowal et al. (2018) employ a 3200-epoch training schedule, using which PGD-7 and IBP respectively attain 34.77% and 24.95% PGD-200-10 accuracy. Mao et al. (2024a) rely instead on a 240-epoch schedule and again tune certified training schemes to maximize certified robustness, reporting 35.93% AA accuracy for PGD-10-3, 36.11% and 36.02% MN-BaB empirical robustness for SABR and MTL-IBP, respectively. In addition, De Bartolomeis et al. (2023) reports better AA accuracy for IBP (Shi et al., 2021) (32.5%), using a 160-epoch schedule, compared to 10-step AutoPGD (Croce & Hein, 2020) (30.2%) on CIFAR-10 for $\epsilon = 8/255$ using a residual network. However, as for their MNIST experiments, the comparison employs different training schedules and optimizers for the various methods, and features an empirically weak adversarial training baseline. We remark that N-FGSM already attains 37.76% average AA accuracy using the same network architecture in table 1, outperforming the results reported above for both adversarial and certified training, and highlighting the importance of strong baselines and appropriate regularization. In fact, differently from Mao et al. (2024a), who adopt $\ell_1$ regularization also for the adversarial training baselines, we use weight decay (with coefficient $5 \times 10^{-5}$), which is more commonly adopted in the adversarial training literature to combat robust overfitting.

### E.2 CO in Certified Training Setups

The strong certified robustness results reported in previous work can be employed to draw conclusions on CO, exploiting the fact that certified accuracy lower bounds the empirical accuracy to any attack. It is easy to conclude that any certified training scheme which lacks an adversarial training component (see §2.3) is trivially immune to CO (Gowal et al., 2018; Shi et al., 2021; Zhang et al., 2020). Nevertheless, these are neither applicable in all training contexts (see §3.2) nor necessarily associated to the best performance (Müller et al., 2023). For $\alpha < 1$ tuned for certified robustness via verifiers based on branch-and-bound, the results reported in De Palma et al. (2024b) imply the absence of CO on the relative CC-IBP, MTL-IBP and Exp-IBP runs. As reported by De Palma et al. (2024b, appendix G.9), this is in spite of the fact that they rely on a randomized attack that always lands on a corner of the perturbation region and was shown to display CO by previous work in a different experimental setup (Wong et al., 2020). However, as highlighted in §4.3, fully demonstrating that the reported results imply the ability of expressive losses to prevent CO requires showing that the underlying one-step attack ($\alpha = 0$) would display CO in the specific experimental setting. We here carry out such investigation, whose results are provided in table 7. In particular, we compare the average AutoAttack accuracy from RS-FGSM $\eta = 10.0\epsilon$, the one-step attack employed in most expressive loss results from De Palma et al. (2024b)[1] with the best AutoAttack accuracy obtained from the published expressive losses checkpoints from De Palma et al. (2024b). Table 7 shows that the one-step attack displays systematic

---

[1]except on CIFAR-10 with $\epsilon = 2/255$, for which De Palma et al. (2024a) use a multi-step attack.

Table 7: CO study on `CNN-7` setups from the certified training literature. We report mean and 95% over 5 runs for the one-step attack used in most expressive loss results from De Palma et al. (2024b), and compare it with the best AutoAttack accuracy across the relative published CC-IBP, MTL-IBP and Exp-IBP checkpoints (De Palma et al., 2024b).

| | MNIST $\epsilon = 0.1$ | MNIST $\epsilon = 0.3$ | CIFAR-10 $\epsilon = 2/255$ | CIFAR-10 $\epsilon = 8/255$ | TinyImageNet $\epsilon = 1/255$ |
|---|---|---|---|---|---|
| Method | AA acc. [%] | AA acc. [%] | AA acc. [%] | AA acc. [%] | AA acc. [%] |
| RS-FGSM $\eta = 10.0\epsilon$ | $83.84 \pm 39.02$ | 0.00 | $74.46 \pm 0.44$ | 0.00 | $28.92 \pm 0.17$ |
| BEST EXPRESSIVE LOSS | 98.48 | 94.02 | 69.33 | 36.50 | 28.46 |

CO on MNIST with $\epsilon = 0.3$ and on CIFAR-10 with $\epsilon = 8/255$, and some signs of CO on MNIST with $\epsilon = 0.1$. In all cases, this is effectively prevented by the expressive losses runs from the literature. Nevertheless, mirroring what is outlined in appendix E.1, we point out that the empirical robustness of literature models is still disappointing if compared with stronger one-step baselines. For CIFAR-10 with $\epsilon = 8/255$, N-FGSM attains 37.76% average AA accuracy on the same network (table 1).

### E.3 Comparison to §5

Differently from the previous literature, this work (§5.1) presents a systematic analysis of the ability of certified training schemes to prevent CO on setups from the single-step adversarial training literature, and (§5.2) extends the study by examining settings where certified training schemes relying on single-step attacks can overcome multi-step attacks.

Owing to the strong sensitivity of CO and of expressive losses (see appendix D.6) to the specific experimental setup, the conclusions in §5.1 do not trivially follow neither from the previous literature, nor from our study in appendix E.2. For instance, figure 8 shows that the occurrence of CO depends on the network architecture even for the same training schedule, dataset, and perturbation radius, and a comparison with table 1 illustrates the effect of changes in the training schedule. Compared to the literature discussed in appendix E.1, §5.2 provides a fair and systematic comparison (relying on the same strong attack suite (Croce & Hein, 2020), which is also used as tuning metric on validation) which includes larger perturbation radii (up to $\epsilon = 24/255$), the harder CIFAR-100 dataset, and a shorter cylic training schedule more common in the adversarial training literature.

Crucially, throughout §5, rather than simply assessing the robustness of certified training schemes deployed to maximize certified robustness, we also tune these algorithms *for* empirical robustness. This can make a difference in practice. For instance, Exp-IBP attains 38.44% average AA accuracy in our experiments for $\epsilon = 8/255$ with $\alpha = 5 \times 10^{-3}$ on CIFAR-10, outperforming N-FGSM and PGD-5 from table 1, the results from table E.2 with $\alpha = 0.5$, and all values reported by Mao et al. (2024a).

