# OpenReview forum: "On Using Certified Training towards Empirical Robustness"
_TMLR — Accepted by TMLR_

### Review · Reviewer_u13t · 2024-12-20

**Summary Of Contributions:**

The paper proposes leveraging various certified training methods combined with single-step attacks to achieve strong empirical robustness, surpassing adversarial training with multi-step attacks. This approach reduces computational overhead and mitigates catastrophic overfitting.

**Audience:**

No

**Claims And Evidence:**

Yes

**Requested Changes:**

1. Provide responses to the weaknesses 1 and 2.
2. Discuss the limitation of lower clean accuracy in the main paper.
3. Add a new graph illustrating how changes in alpha affect empirical robustness.

**Strengths And Weaknesses:**

### Strengths:
1. The paper is well-written and includes a thorough summary of related works.
2. Results are presented clearly across various experimental settings.

### Weaknesses and Questions:
1. The reviewer is concerned about whether the paper provides sufficient new insights for the community’s interest. As discussed in Section 4.3, it is well understood that certified training tends to exhibit strong empirical robustness under certain conditions, such as for small networks or large adversarial perturbation budgets. In both cases, the certification becomes tighter, and the model behaves more linearly. The training objective proposed in this paper essentially combines standard adversarial loss with a certified objective. Consequently, compared to multi-step adversarial training, the proposed method is effective when either of these two conditions is met.
Furthermore, the contribution of Exp-IBP in preventing catastrophic overfitting (CO) appears to be implicitly covered by the findings of De Palma et al. (2024b). They demonstrated that Exp-IBP, when used with one-step adversarial examples, can achieve strong certified robustness (a lower bound of empirical robustness), thereby mitigating CO.
2. The paper emphasizes that “certified and adversarial training have so far evolved as two orthogonal lines of work with distinct objectives… This belief is consolidated by a wide and persisting performance gap between the best-reported certified and empirical reobustness for a given setup.” However, this claim feels overstated. Many certified robustness studies also examine empirical robustness; adversarial training inherently targets at empirical robustness and thus, only empirically robustness is evaluated (the certified robustness performance will be poor). The gap between certified and empirical robustness is expected, as certified robustness is constrained by the certification method and serves as a lower bound, while empirical robustness is an upper bound based on empirical evaluations. Better certification and training, and better empirical attacks will make the gap smaller. The reviewer feels like the Orthogonality is overemphasized here.
3. Figure 5 – Evaluation with AA:Could the authors evaluate empirical robustness using AutoAttack (AA)? The community recognizes that even PGD-7 can suffer from gradient obfuscation, making AA a more reliable benchmark.
4. Figure 6: The color representation of PGD-10 should be consistent across the two subfigures to aid interpretation.
5. All methods utilizing a certified objective show lower clean accuracy compared to adversarial training, as indicated in Appendix D.4. This is a significant limitation and warrants discussion in the main paper.
6. Table 3.3 highlights significant variations in the alpha parameter across different settings. While using certified objectives may reduce computational overhead, the tuning process for alpha appears complex. This complexity might negate the claimed time savings compared to direct adversarial training. Results examining the sensitivity of robustness outcomes to alpha are needed, especially given the frequent mention of "explicit tuning" in the paper.

---

> ### Author Response · Authors · 2025-02-09
>
> We sincerely thank the reviewer for the valuable comments, the detailed feedback, and for the positive evaluation concerning the writing and the experimental work.
> We performed the requested changes in the text (highlighted in blue), and thank the reviewer for helping us improve the quality of the paper.
> We will now reply to the raised points individually (in two separate replies owing to character limits).
>
> 1) As the reviewer rightly pointed out, section 4.3 clearly acknowledges that some previous works [Gowal et al., 2018; Mao et al. 2024] have anecdotally reported strong empirical robustness for certified training schemes.
> Nevertheless, as detailed in appendix E, *their conclusions lack generality*: they either pertain to MNIST (where the best-reported certified accuracies are extremely large), or to training CNN-7 on CIFAR-10 for $\epsilon=\frac{8}{255}$ against severely-undertuned multi-step baselines (in our experimental setup their certified training results are outperformed by a single-step baseline, on the same network).
> Given the specificities of certified training schemes, which are typically applied on shallow networks and over long training schedules, *demonstrating that these conclusions can generalise to setups of interest in the single-step adversarial training literature it is hence far from obvious*.
> For instance, Figure 6 shows that, on a single-step literature setup featuring a much deeper `PreactResnet18` and a much shorter training schedule, certified training techniques outperform PGD-5 on CIFAR-10 for $\epsilon=\frac{24}{255}$ while even incurring a smaller runtime overhead.
> Amongst other things, we then systematically study how these conclusions are affected by changes in network architecture or training schedule.
> Furthermore, we agree that literature results offer partial insights into the relationship between CO and certified training schemes.
> In particular, the certified training results from [De Palma et al., 2024b] do not display CO in spite of the use of weak single-step attacks for the adversarial loss component, as stated in their appendix G.9.
> However, the occurrence of CO heavily depends on the specific training setup (see for instance figure 8). It is hence unclear whether (i) setting $\alpha=0$ would indeed display CO (hence implying that certified training can prevent it) in the employed experimental setting, and (ii) whether these findings would generalise to settings from the single-step adversarial training literature.
> We particularly thank the reviewer for this comment: we have edited the text to acknowledge and discuss these points at length in section 4.3 and appendix E.
> Notably, appendix E now presents additional experiments which address point (i), hence extending our contributions.
> Finally, in contrast to previous work, we tune certified training algorithms *for* empirical robustness: as discussed in appendix E, this does make a difference in practice.
> In conclusion, differently from literature results, our study of the utility of certified training techniques towards empirical robustness is *targeted* (including relevant setups from the adversarial training literature), *intentional* (empirical robustness is explored as a primary goal, not as a by-product), and *systematic* (studies the joint effect of architectures, training schedules, and of datasets).
> As a result, to paraphrase the TMLR acceptance criteria and as confirmed by the other reviewers, we strongly believe our findings and analysis to be of interest to the adversarial ML community.
>
> 2) Thank you for your feedback, which we incorporated by editing the relevant introduction paragraph. What we wanted to convey is that certified training schemes are typically understood to improve verifiability *at the expense of* empirical robustness, which is far from being their primary goal.
> This is reflected by the fact that many recent papers introducing novel certified algorithms do not even report their empirical robustness within their main results [Müller et al., 2023; Mao et al., 2023; De Palma et al., 2024b].
> In fact, for a given benchmark (say, CIFAR-10 $\epsilon=\frac{8}{255}$), the gap between the best-reported certified accuracy *across networks and algorithms* (40.39% from [Zhang et al., 2022a], which has 41.5% PGD-100 accuracy) and the best-report empirical accuracy (to AutoAttack) *across networks and algorithms* (at least ~60% according to the RobustBench leaderboard) is significant.
> The goal of our work is to instead study the behaviour of certified training schemes *when empirical robustness is employed as their primary goal*.
>
> 3) We have updated Figure 5 tuning with AA evaluation during training, confirming that Exp-IBP, MTL-IBP and ForwAbs can prevent CO in this setting.
>
> 4) We thank the reviewer for pointing this out and have updated the figure for colour consistency.

---

> > ### Author Response · Authors · 2025-02-09
> >
> > 5) The effect on clean performance of the runs in sections 5.1 and 5.2, which were tuned to exclusively maximise empirical robustness, is now prominently featured in 5.1.2 and 5.2.1. Nevertheless, we stress that better trade-offs between robustness and standard performance may be obtained through alternative tuning criteria (see reply to point 6 below).
> >
> > 6) Thanks for pointing this out: we added a study of the effect of the $\alpha$ and $\lambda$ coefficients on the performance of Exp-IBP and ForwAbs, respectively, as section 5.3.
> > The sensitivity of each method depends on the goal and on the experimental setup.
> > While CO can be prevented by a wide range of coefficients on all the setups considered within the section 5.3 experiments, maximising AutoAttack accuracy (or another notion of empirical robustness, such as PGD-50 accuracy) requires more fine-tuning, especially on harder datasets and larger networks.
> > Importantly, the experiment demonstrates that the standard performance cost compared to N-FGSM may be significantly improved (to the point of becoming negligible on CIFAR-10 $\epsilon=\frac{20}{255}$) by taking clean accuracy into account at tuning time.

---

### Review · Reviewer_PdVS · 2024-12-20

**Summary Of Contributions:**

1. Main Insight of paper - Multi-step adversarial attacks tighten network over-approximations, reducing IBP (Interval Bound Propagation) loss, unlike single-step attacks, which are prone to catastrophic overfitting. The authors hypothesize that certified training can enhance robustness for single-step adversarial training methods.

2. Empirical Findings:
    * Certified training with Exp-IBP prevents catastrophic overfitting.
    * On shallower networks, IBP outperforms multi-step adversarial baselines like PGD-10 under specific conditions.
    * A novel regularizer, ForwAbs, achieves robustness similar to Exp-IBP while being computationally cheaper.

3. The introduction of ForwAbs as a proxy for IBP loss demonstrates to balance compute efficiency and performance.

**Audience:**

Yes

**Claims And Evidence:**

Yes

**Requested Changes:**

1. Improve Clarity in Section 5:

    a. Split results into distinct experiments, the current 5.1 and 5.2 sections are providing multiple results and many takeaways and is very hard to parse for the reader.

    b. At the end of each experiment consider summarizing key takeaways.

2. Reorganize Efficiency Discussion:

    a. The main claim to fame for ForwAbs is efficiency. Relocate runtime analyses from Appendix D.2 to the main body.

    b. Include a trade-off analysis plot showing compute costs (runtime on x-axis) versus robustness (AA accuracy/IBP accuracy on y-axis) to emphasize where the compute-robustness trade-offs lie for various methods.

**Strengths And Weaknesses:**

Strengths

* Key Insight Validation: The study identifies how multi-step attacks inherently promote robustness, providing a new perspective on certified training's utility for empirical defenses.
*  Technique: ForwAbs introduces a computationally efficient alternative, reducing overhead without significantly compromising IBP effectiveness.


Weaknesses
*  Limited Scope of Activation Functions: The techniques and bounds proposed are optimized for ReLU activation, limiting applicability to networks using other activation types.
* Unclear Messages in Section 5.1: The narrative of key results lacks clarity, making it difficult for readers to distill the main takeaways.
* Efficiency Claims in Appendix: Core claims of computational efficiency for the proposed methods are relegated to the appendix, undermining their significance.

---

> ### Author Response · Authors · 2025-02-09
>
> We sincerely thank the reviewer for their time, valuable comments, and for appreciating the paper's insights.
> The requested changes have been integrated into the manuscript (in blue): thank you for helping us improve the quality of the presentation and of the paper.
> We now summarise the changes and reply to the individual weaknesses pointed out in the review.
>
> ### Limited Scope of Activation Functions
>
> ReLU networks are indeed the main focus of our investigation, in line with both the certified training and single-step adversarial training literatures [Gowal et al., 2018; Müller et al., 2023; Mao et al., 2023; De Palma et al. 2024b; Wong et al., 2020; de Jorge et al., 2022; Rocamora et al., 2024].
> Nevertheless, we believe that certified training techniques may well be applicable for empirical robustness beyond ReLU networks, and thank the reviewer for the comment.
> In order to provide evidence in this direction, we carried out a SoftPlus experiment using a modified `PreActResNet18` where each ReLU is replaced by a SoftPlus (which displays a significant qualitative difference: it is not piecewise-linear).
> We leave the ForwAbs implementation unchanged, and use the procedure from section 2.3.1 (which applies to any monotonic activation function) to compute the IBP bounds.
> Appendix D.8 shows that (i) FGSM suffers from CO on the SoftPlus network on CIFAR10 and CIFAR100 at $\epsilon=\frac{24}{255}$ and that (ii) Exp-IBP, MTL-IBP and ForwAbs can prevent it, hence demonstrating the applicability of certified training techniques towards empirical robustness beyond ReLU networks.
>
>
> ### Clarity in Section 5
>
> We have now divided subsections 5.1 and 5.2 into a subsection per experiment (two subsections each), where the descriptions of the experimental setting and of the conclusions are now clearly separated.
> Furthermore, in order to improve readability, we have summarised (where it was not already the case) the main takeaways for each of the relative experiments in the respective caption.
> Thank you for helping us improve the quality of the presentation.
>
> ### Efficiency Discussion
>
> Thank you for your comment: we agree that runtime costs should have featured more prominently in the paper.
> Figure 6 now shows a measurement of the training overhead of each of the presented algorithms beside their experimental performance.
> In addition, appendix D.2 now displays the requested trade-off analysis plot for three different setups (figure 18).
> Across the three setups, both ForwAbs and Exp-IBP consistently outperform PGD-5 while reducing its runtime, and the strong compute-robustness trade-offs of ForwAbs are particularly visible.

---

### Review · Reviewer_i76a · 2025-01-26

**Summary Of Contributions:**

The authors explore the utility of certified training schemes to boost empirical robustness in adversarial training. They find a connection between the phenomenon of catastrophic forgetting observed in fast adversarial training algorithms and local linearity of network gradients. Specifically, they show that IBP losses of multi-step adversarial trained networks are considerably lower compared to those trained with single-step attacks, which motivates them to explore combining single-step methods with IBP.
1) They find that combining single-step adv. training with state-of-the-art IBP losses is competitive to multi-step adversarial training with sufficient hyperparameter tuning
2) They investigate shallow networks in more detail and demonstrate that long training schedules and IBP can outperform standard adv. training in some cases.
3) They propose an efficient to compute alternative loss to IBP, which prevents catastrophic overfitting and enables fast and robust single-step adv. training.

**Audience:**

Yes

**Claims And Evidence:**

Yes

**Requested Changes:**

- Add a study investigating best vs. last robust accuracy using a validation set
- Could the authors clarify on the concern stated in W3?

**Strengths And Weaknesses:**

**Strengths**
- To the best of my knowledege there is only a small number of works comparing and combining empirical and certified robustification approaches.
- The authors conduct an extensive study on combining IBP with adversarial training and demonstrate cases where certified robustness approaches can achieve competitive empirical robustness. This finding may motivate further research into this area, such as investigations on the empirical robustness of Lipschitz-constant architectures or the combination of Lipschitz-bound and empirical methods.
- The authors provide further insights into catastrophic overfitting by relating it to the IBP objective
- For all claims sufficient experimental evidence is provided

**Weaknesses**

W1: While I personally think that studies conducted on cifar10 are sufficient for this kind of research it would have been interesting to see if the proposed approach follows the same scaling trends as standard adv. training procedures concerning model size and dataset size. The biggest improvements in robustness have been made by scaling dataset size.

W2: Could the authors provide an experiment investigating the difference between the maximum validation robust accuracy and the last validation robust accuracy to get further insights on the ability of IBP losses to reduce CO?

W3: I understood that the authors did not try to push sota robustness in their work. However, some robustness values for larger epsilon balls appear to be very small and it is a bit unclear if an improvement in this regime is important?

---

> ### Author Response · Authors · 2025-02-09
>
> We sincerely thank the reviewer for their time and detailed comments.
> We are particularly glad that the reviewer found our study to be extensive and appropriately backed by experimental evidence, and for stating that it may motivate further research in the area: we totally agree that investigating the behaviour of low-Lipschitz architectures is an exciting avenue for future work.
> We now address each of the reported weaknesses individually (requested changes in the text of the submission are highlighted in blue).
>
> > W1
>
> Thank you for your comment: we agree that scaling trends are interesting, and have added a new appendix (D.6.1) to provide an explicit analysis of the effect of model architecture on the performance of both multi-step adversarial training and of certified training techniques.
> While the empirical robustness of ForwAbs, PGD-5 and PGD-10 is maximised on `CNN-7` (where capacity is used for width rather than depth), Exp-IBP performs the best on the shallower `CNN-5`, highlighting a markedly different behaviour for certified training techniques that employ the IBP loss.
> We ascribe this (see the newly-added appendix D.6.2) to the magnitude of the IBP bounds at initialisation: smaller networks reduce the compositional effect associated with the composition of each layer's over-approximation.
> These results demonstrate significant qualitative differences between the scaling trends of multi-step adversarial training and of techniques based on IBP.
> For what concerns the effect of additional data, we point out that enabling expressive losses to benefit from it is an active area of research within the certified training literature: see negative results in appendix H of [1].
> As a result, we believe that carrying out such analysis is beyond the scope of this submission (albeit it is undeniably an exciting direction for future work).
>
> [1] On the Scalability of Certified Adversarial Robustness with Generated Data, Altstidl et al., NeurIPS 2024
>
> > W2
>
> The newly-added appendix D.5 presents a comparison of the performance of Exp-IBP, ForwAbs and N-FGSM on their final epoch against their "best" epoch, selected according to the PGD-50 accuracy on a validation set of 1000 images.
> On the two considered setups (CIFAR-10 $\epsilon=\frac{20}{255}$ and CIFAR-100 $\epsilon=\frac{24}{255}$, where N-FGSM displays the most visible signs of CO), both Exp-IBP and ForwAbs significantly reduce the variability between the final and the selected models, further demonstrating their ability to prevent CO when employed on top of N-FGSM.
> We thank the reviewer for leading to this addition.
>
> > W3
>
> Large perturbation radii are typically considered to be settings of interest in the single-step adversarial training literature.
> For instance, [Rocamora et al., 2024] highlight that even the otherwise effective N-FGSM suffers from what they call "delayed" CO: it is immune to CO on smaller perturbation radii, but displays the failure mode when $\epsilon$ is increased.
> Testing CO prevention schemes (such as, as we show, certified training techniques) on large $\epsilon$ values is hence extremely important for a thorough evaluation of their capabilities.
> In their appendix B, [Rocamora et al., 2024] show that, while large perturbation radii are visually perceptible, they do not affect a human's oracle classification, even for $\epsilon=\frac{26}{255}$, the larger radius they consider.
> Robustness against these perturbation magnitudes is hence desirable.
> Other single-step adversarial training works [Lin et al., 2023; Lin et al. 2024] report results until $\epsilon=\frac{32}{255}$, further showing the relevance of large-$\epsilon$ settings.

---

> > ### Comment · Reviewer_i76a · 2025-02-11
> > **Thanks for the clarifications**
> >
> > I want to thank the authors for the information they provided.
> >
> > W1: I agree that scaling certified training schemes can be viewed as outside the scope of this work. My other concern has been addressed.
> >
> > W2: Thanks for conducting the experiment. I believe this further supports your claim that the proposed approach addresses CO.
> >
> > W3: Thanks for the reference to related literature. Given prior interest in this threat model I agree that it may be relevant to other researchers of the TMLR community.
> >
> > Overall my concerns were sufficiently addressed by the rebuttal and the updated manuscript and I would recommend to accept the paper.

---

### Review · Reviewer_J5KY · 2025-01-26

**Summary Of Contributions:**

This paper shows that certified training methods like Exp-IBP and the proposed ForwAbs regularizer prevent catastrophic overfitting and reduce computational costs by approximating certified bounds efficiently in single-step adversarial training, matching multi-step methods in empirical robustness on CIFAR-10 for perturbations.

**Audience:**

Yes

**Claims And Evidence:**

Yes

**Requested Changes:**

- Address the questions raised in the weaknesses section.

**Strengths And Weaknesses:**

> Strengths:

- The paper addresses a critical challenge in adversarial robustness—closing the gap between empirical and certified robustness. This is both timely and relevant, given the need for robustness guarantees in safety-critical applications.
- Exp-IBP and ForwAbs effectively mitigate CO in single-step adversarial training. Introducing ForwAbs for improving empirical robustness with reduced overhead.
- The paper is well articulated and structured.

> Weakness:

- The claim that Exp-IBP can outperform multi-step adversarial training baselines is valid only under carefully tuned setups and for specific datasets (CIFAR10), which diminishes its generality.
- The reliance on extensive hyperparameter tuning (α for Exp-IBP and λ for ForwAbs) makes the method less practical for broader adoption, especially in large-scale settings. Also, it will be interesting to add sensitivity studies to analyze the effect of α and λ  over a range on both accuracy under different perturbation levels.
- Lack of comparison against stronger multi-step adversarial training methods like TRADES, MART, or more recent AutoAttacks.
-  Shallow CNNs (CNN-5/7) benefit most, while deeper models (PreActResNet18) show mixed results. The paper does not investigate why deeper networks struggle.

---

> ### Author Response · Authors · 2025-02-09
>
> We sincerely thank the reviewer for their time and valuable feedback, and we are glad that our work was found to be timely, relevant and well-structured.
> As stated in the general response, we would like to emphasise that the goal of our work is not to present a new state-of-the-art in adversarial robustness, but rather to investigate and shed light on the (so-far neglected) utility of certified training techniques towards empirical robustness, paving the way for further research in the area.
> We performed the requested changes in the submission text (highlighted in blue), and will now reply to the specific questions raised by the reviewer (in two separate replies owing to character limits).
>
> ### Generality of the results
>
> Certified training schemes are typically understood to improve verifiability *at the expense of* empirical robustness, which is far from being their primary goal.
> As a result, demonstrating that, differently from the conventional understanding, these techniques can be successfully leveraged towards empirical robustness is arguably already of great interest to the adversarial machine learning community.
> The fact that certified training techniques (Exp-IBP), when applied on top of a single-step attack which suffers from CO, can perform competitively with multi-step adversarial training baselines in a setup of proven interest to the single-step adversarial training literature while reducing their runtime (Figure 6) is hence arguably even more remarkable.
> Nevertheless, we agree that the relative utility of certified training schemes is maximised under specific experimental conditions: indeed, determining them and studying them is exactly the subject of section 5.2.
> Our work pinpoints the potential and limitations of certified training algorithms applied for empirical robustness, constituting a crucial step towards future advances in the area.
>
> ### Hyper-parameters / Sensitivity
>
> Section 5.3 presents a newly-added study of the effect of the $\alpha$ and $\lambda$ coefficients on the performance of Exp-IBP and ForwAbs, respectively: we thank the reviewer for the suggestion, which helped us improve the quality of our submission.
> Figures 10 and 11 show that the amount of tuning required for each method depends on the goal and on the experimental setup.
> If CO prevention is the primary goal, then this is achieved across a wide range of coefficients.
> Maximising empirical robustness requires more fine-tuning, especially on harder datasets (CIFAR-100).
> Nevertheless, we would like to point out that it is far from uncommon for algorithms for CO prevention to present an additional hyper-parameter.
> This is for instance the case for GradAlign [Andriushchenko and Flammarion, 2020] and for the state-of-the-art ELLE [Rocamora et al., 2024], which relies on a significant amount of fine-tuning as described in their appendix B.2.
> Other works [Lin et al, 2023; Lin et al. 2024] introduce more than a hyper-parameter at once, further increasing the tuning overhead.
>
> ### Stronger multi-step adversarial methods
>
> Our work focuses on the application of certified training techniques on top of single-step attacks and aims to showcase their utility towards improving empirical robustness in this context.
> Owing to the vulnerability of single-step adversarial training to CO, multi-step PGD training (usually, PGD-10) is typically employed as a strong baseline, whose performance is typically unmatched in the relevant single-step literature [de Jorge et al., 2022; Rocamora et al., 2024].
> Because of this, and given that PGD-10 outperforms certified training in the vast majority of our experiments, we omitted stronger baselines from the submission.
> Nevertheless, following the reviewer's suggestion, we ran MART and TRADES on the two setups where, in our experiments, PGD-10 is outperformed by certified training schemes (IBP): CIFAR 10 with $\epsilon \in \{\frac{16}{255},\frac{24}{255}\}$ from the long schedule setup of Table 1.
> The relative results, which compare unfavourably to PGD-10 from Table 1, are reported in the following table:
>
> | Method   |   Epsilon | AAev_aa_ok   | AAev_pgd_ok   | AAev_nat_ok   |
> |:---------|----------:|:-------------|:--------------|:--------------|
> | MART     |        16 | 19.51 ± 0.56 | 32.29 ± 0.58  | 52.95 ± 0.46  |
> | TRADES   |        16 | 13.29 ± 0.44 | 17.63 ± 0.58  | 35.36 ± 0.72  |
> | MART     |        24 | 10.34 ± 0.66 | 23.30 ± 0.34  | 34.74 ± 0.71  |
> | TRADES   |        24 | 8.00 ± 0.49  | 12.53 ± 0.49  | 33.20 ± 0.52  |
>
> We were ourselves surprised by the disappointing performance of both methods, in spite of extensive tuning of the respective coefficients (denoted $\beta > 0$) in the following range: $\beta \in \{1, 5, 10, 20, 25, 100, 500, 1000 \}$.
> We would be happy to try any specific setting or hyper-parameter for which the reviewer believes a better performance should be expected.
>
> [1] Improving Adversarial Robustness Requires Revisiting Misclassified Examples. Wang et al., ICLR 2020.

---

> > ### Author Response · Authors · 2025-02-09
> >
> > ### Why deeper networks struggle
> >
> > Thank you for your comment.
> > In order to provide insights on the relative performance of certified training techniques on the `CNN` and `PreActResNet18` architectures, appendix D.6.2 of the revised submission provides a table with the relative IBP loss at initialisation (computed on the CIFAR-10 training set for $\epsilon=\frac{24}{255}$), which we report here for convenience:
> >
> > | Architecture       | IBP loss                |
> > |--------------------|-------------------------|
> > | PreActResNet18     | $2.20 \pm 0.06 \times 10^{16}$ |
> > | CNN-7              | $8.67 \pm 0.18 \times 10^{5}$  |
> > | CNN-5              | $1.11 \pm 0.01 \times 10^{4}$  |
> >
> > As expected from the certified training literature, the IBP loss explodes with depth on `PreActResNet18`, which displays a loss more than 10 orders of magnitude greater than on the `CNN` models.
> > As a result, a greater share of the network's capacity is devoted to meaningfully decreasing the IBP bounds, hurting the final network robustness (this is further reflected in the trends described in appendix D.6.1).
> > We believe that major improvements on certified training algorithms will be required to enable effective scaling onto deeper networks.

---

### Author Response · Authors · 2025-02-09
**General Response**

We sincerely thank all the reviewers for their valuable comments, suggestions, and requested changes: these have all helped us improve the quality of the submission.

### Interest of the findings

We are glad that reviewers `PdVS` ("*a new perspective on certified training's utility for empirical defenses*"), `i76a` ("*This finding may motivate further research into this area*"), and `J5KY` ("*The paper addresses a critical challenge in adversarial robustness*") found our work to be insightful and of interest to the community.
Indeed, we believe our work to shed new light on the utility (and persisting limitations) of using certified training algorithms towards empirical robustness, paving the way for further developments in the area.

### Provided evidence

As supported by reviewers `u13t` ("*Results are presented clearly across various experimental settings*") and `i76a` ("*conduct an extensive study on combining IBP with adversarial training and demonstrate cases where certified robustness approaches can achieve competitive empirical robustness*", "*For all claims sufficient experimental evidence is provided*"), we provide an extensive and comprehensive empirical evaluation to support our claims.
For instance, in the words of reviewer `J5KY`, that "*Exp-IBP and ForwAbs effectively mitigate CO in single-step adversarial training*".
In addition to studying the effect of various network architectures, perturbation sizes, and training schedules, the relevance of our conclusions is strengthened by their validity on `PreActResNet18` settings popular in the single-step adversarial training literature [Andriushchenko and Flammarion, 2020; de Jorge et al., 2022; Rocamora et al., 2024], where CO is prevented and PGD-5 is outperformed for CIFAR-10 $\epsilon=\frac{24}{255}$.
Indeed, we emphasize that our goal is not to propose a new state-of-the-art in adversarial training: our claims focus on the ability of Exp-IBP and ForwAbs to prevent CO across setups, and on their competitiveness with strong multi-step baselines under specific settings (larger epsilons, shallower networks, longer schedules).

### Main revision improvements

We now list the main improvements in the revised submission (highlighted in blue in the submission text):

- As kindly suggested by reviewers `u13t` and `J5KY`, section 5.3 now presents a study on the effect of the $\alpha$ and $\lambda$ coefficients on the performance of Exp-IBP and ForwAbs.
On top of showing that CO can be prevented for a wide range of coefficients, it highlights that very different performance profiles can be obtained depending on the tuning criterion.
In particular, clean accuracy can be taken into account within the tuning to produce stronger robustness-accuracy trade-offs (this was out of the scope of sections 5.1 and 5.2), demonstrating the versatility of certified training algorithms in the context of empirical robustness.

- Appendix D.8 now shows that Exp-IBP, MTL-IBP and ForwAbs can prevent CO also on SoftPlus activations, addressing a comment by reviewer `PdVS`.
Owing to the fact that SoftPlus is not piecewise-linear, this demonstrates the applicability of certified training techniques for empirical robustness beyond ReLU networks, which are the usual focus of the single-step adversarial training literature.

- We updated our positioning within the relevant literature.
First, addressing a comment by reviewer `u13t`, section 4.3 now lists the partial insights that can be drawn on CO from previous certified training work, with appendix E.2 presenting an explicit experimental study that deepens some of these and complements the results in the main paper.
Second, we acknowledge a pre-existing work [Boopathy et al., 2021], which we saw while the submission was under review, that presents a certified training algorithm, evaluated and designed for certified accuracy only, whose cost is a single network evaluation.

- As requested by reviewer `PdVS`, we featured computational efficiency, which is an advantage of the considered certified training techniques compared to multi-step adversarial training, more prominently in the main body of the paper, and extended the analysis in appendix D.2 to better illustrate compute-robustness trade-offs. We have furthermore improved the organisation of section 5, summarising the main takeaways from the experiments in sections 5.1 and 5.2 in each experiment's caption.

- Addressing comments from reviewers `J5KY` and `i76a`, appendix D.6 presents additional insights on the role of model architecture on performance. In particular, we link the strong performance of certified training techniques on smaller networks to their significantly smaller IBP losses at initialisation, further explaining the presented results.

### Invididual responses

We thank again all reviewers for their valuable feedback, and refer to the individual responses for reviewer-specific discussions. We will be happy to engage in further discussions if needed.

---

### Decision · Action_Editor_Grg3 · 2025-02-24

**Recommendation:** Accept as is

**Comment:**

All reviewers recommend accept. Reviewer u13t (Leaning Accept) recommends leaning towards accepting after rebuttal with the added graph. Reviewer i76a (Accept) recommends acceptance after the rebuttal sufficiently addresses original concerns. Reviewer PdVS (Leaning Accept) addresses the limitations but has no further reasons to reject the paper.

**Audience:**

At least some individuals in TMLR's audience would be interested in knowing the findings of this paper. Reviewer PdVS points out that the paper is "providing a new perspective on certified training's utility for empirical defenses," suggesting a novel approach. Additionally, Reviewer i76a suggests this "finding may motivate further research into this area," indicating that it contributes to the conversation.

**Claims And Evidence:**

The claims made in the submission are generally supported by evidence. Reviewer u13t found that "Results are presented clearly across various experimental settings," and Reviewer i76a notes that "For all claims sufficient experimental evidence is provided"